# ONLINE CONTINUAL LEARNING FOR PROGRESSIVE DISTRIBUTION SHIFT (OCL-PDS): A PRACTITIONER'S PERSPECTIVE

## ABSTRACT

We introduce the novel OCL-PDS problem - Online Continual Learning for Progressive Distribution Shift. PDS refers to the subtle, gradual, and continuous distribution shift that widely exists in modern deep learning applications. It is widely observed in industry that PDS can cause significant performance drop. While previous work in continual learning and domain adaptation addresses this problem to some extent, our investigations from the practitioner's perspective reveal flawed assumptions that limit their applicability on daily challenges faced in real-world scenarios, and this work aims to close the gap between academic research and industry. For this new problem, we build 4 new benchmarks from the Wilds dataset (Koh et al., 2021), and implement 12 algorithms and baselines including both supervised and semi-supervised methods, which we test extensively on the new benchmarks. We hope that this work can provide practitioners with tools to better handle realistic PDS, and help scientists design better OCL algorithms.

## 1 INTRODUCTION

In most modern deep learning applications, the input data undergoes a continual distribution shift over time. For example, consider a satellite image classification task as illustrated in Figure 1a. In this task, the input data distribution changes with time due to the changes in landscape, and camera updates which can lead to higher image resolutions and wider color bands. Similarly, in a toxic language detection task on social media illustrated in Figure 1b, the distribution shift can be caused by a shift in trends and hot topics (many people post about hot topics like BLM (Wikipedia contributors, 2022a) and Roe v. Wade (Wikipedia contributors, 2022b) on social media), or a shift in language use (Röttger & Pierrehumbert, 2021; Luu et al., 2022). Such distribution shift can cause significant performance drop in deep models, a widely observed phenomenon known as *model drift*.

A critical problem for practitioners, therefore, is how to deal with what we term *progressive distribution shift (PDS)*, defined as **the subtle, gradual, and continuous distribution shift that widely exists in modern deep learning applications**. In this work, we explore handling PDS with *online continual learning (OCL)*, where **the learner collects, learns, and is evaluated on online samples from a continually changing data distribution**. In Section 2, we formulate the OCL-PDS problem.

The OCL-PDS problem is closely related to two research areas: *domain adaptation (DA)* and *continual learning (CL)*, in which there is a rich body of academic work. However, through a literature review and our conversations with practitioners, we find that there still remains a gap between the settings widely used in academic work and in real industrial applications. To close this gap, we commit ourselves to **thinking from a practitioner's perspective**, which is the core spirit of this work. Our primary goal is to build tools for investigating the real issues practitioners are facing in their day-to-day work. To achieve this goal, we challenge the prevailing assumptions in previous work, and propose three important modifications to the conventional DA and CL problem settings:

1. **Task-free:** One point conventional DA and CL settings have in common is assuming clear boundaries between distinct domains (or tasks), but practitioners rarely apply the same model to very different domains in industry. In contrast, OCL studies the *task-free* CL setting (Aljundi et al., 2019b) where there is no clear boundary, and the distribution shift is continuous. Moreover, in OCL both training and evaluation are online, unlike previous task-free CL settings with offline evaluation, which is not as realistic in a "lifelong" setting.

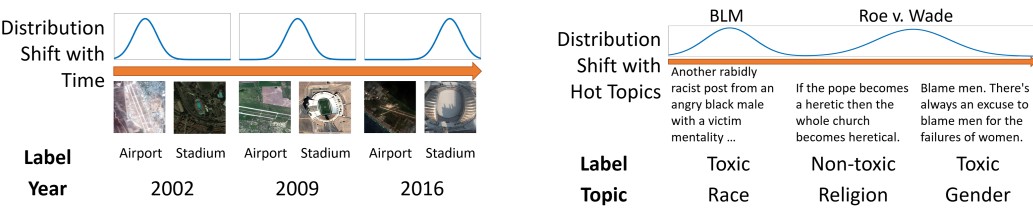

(a) FMoW-WPDS benchmark.        (b) CivilComments-WPDS benchmark.

Figure 1: FMoW-WPDS and CivilComments-WPDS benchmarks which we build in this work.

2. **Forgetting is allowed:** Avoiding *catastrophic forgetting* is a huge topic in CL, which usually requires no forgetting on *all* tasks. However, remembering *everything* is actually impractical, infeasible and potentially harmful, so OCL-PDS only requires remembering recent knowledge and important knowledge which is described by a *regression set* (Sec. 2.2).

3. **Infinite storage:** Previous work in CL usually assumes a limited storage (buffer) size. However, storage is not the most pressing bottleneck in most industrial applications. Thus, in OCL-PDS, we assume an infinitely large storage where all historical samples can be stored. However, the learner cannot replay all samples because it would be too inefficient.

To demonstrate the novelty and practicality of the OCL-PDS problem, in Section 2.3 we will discuss related work, compare OCL-PDS with common and similar settings used in previous work, and elaborate on the reasons why we believe these three key modifications align OCL-PDS more closely with industrial applications and practitioners' pain points. A more thorough literature review can be found in Appendix A. Then, in Section 3, we will build 4 new benchmarks for OCL-PDS, including both vision and language tasks. When building these benchmarks, we make every effort to make sure that they can reflect real PDS scenarios that practitioners need to deal with.

In Section 4, we will explore OCL algorithms and how to combine them with semi-supervised learning (SSL) as unlabeled data is very common in practice. In total, we implement 12 supervised and semi-supervised OCL algorithms and baselines adapted to OCL-PDS, which we test extensively on our benchmarks in Section 5. Our key observations in these experiments include: (i) A task-dependent relationship between learning and remembering; (ii) Some existing methods have low performances on regression tasks; (iii) SSL helps improve online performance, but it requires a critical virtual update step. Finally, in Section 6 we discuss remaining problems and limitations.

**Contributions.** Our contributions in this work include: (i) Introducing the novel OCL-PDS problem which more closely aligns with practitioners' needs; (ii) Releasing 4 new benchmarks for this novel setting; (iii) Adapting and implementing 12 OCL algorithms and baselines, including both supervised and semi-supervised, for OCL-PDS; (iv) Comparing these algorithms and baselines with extensive experiments, which leads to a number of key observations. Overall, we believe that this work is an important step in closing the gap between academic research and industry, and we hope that this work can inspire more practitioners and researchers to investigate and dive deep into real-world PDS. To this end, we release our benchmarks and algorithms, which are easy to use and we hope can help boost the development of OCL algorithms for handling PDS.

## 2 THE OCL-PDS PROBLEM

### 2.1 PROBLEM FORMULATION

We have a stream of online data $S_0, S_1, \cdots$, where each $S_t$ is a batch of *i.i.d.* samples from distribution $D_t$ that changes with time $t$ continuously, for which we assume that $\text{Div}(D_t \parallel D_{t+1}) < \rho$ for all $t$ for some **divergence function** Div. Online Continual Learning (OCL) goes as follows:

- At $t = 0$, receive a *labeled* training set $S_0$, on which train the initial model $f_0$

- For $t = 1, 2, \cdots, T, \cdots$ do

    1. Data collection: Receive a new *unlabeled* **data batch** $S_t = \{(x_t^{(i)}, y_t^{(i)})\}_{i=1}^{n_t} \overset{\text{i.i.d.}}{\sim} D_t$.
    2. Evaluation: Predict on $S_t$ with the current model $f_{t-1}$, and get some **feedback**
    3. Fine-tuning: Update the model $f_{t-1} \to f_t$ with **all previous information**

**Evaluation metrics.** An OCL algorithm is used for fine-tuning and is evaluated by three metrics:

1. **Online performance:** Denote the performance of $f_s$ on $S_t$ by $A_s^t$. The online performance at time $t$ (as computed in Step 2 - Evaluation) is $A_{t-1}^t$, and the *average online performance* before horizon $T$ is defined as $(A_0^1 + \cdots + A_{T-1}^T)/T$.

2. **Knowledge retention:** Unlike conventional CL that requires no forgetting on all tasks, in OCL-PDS the model only needs to remember two types of knowledge: **recent knowledge** and important knowledge. For a certain *recent time window* $w$, the *recent performance* is defined as $(A_{t-1}^{t-w} + \cdots + A_{t-1}^{t-1})/w$. The important data is described by a **regression set**, and the *regression set performance* is the model's performance on this set.

3. **Training efficiency:** This is measured by the *average runtime* of the fine-tuning step, which is very important for this online setting where the OCL algorithm is run for many times.

## 2.2 DETAILS

**Divergence function.** For distributions $P$ and $Q$, $\mathrm{Div}(P \parallel Q)$ is the divergence *from $P$ to $Q$*, and can be different from $\mathrm{Div}(Q \parallel P)$. According to our reasoning in Appendix B, an ideal divergence function for OCL-PDS should be *asymmetric and bounded*, so we cannot use popular functions such as total variation, Wasserstein distance, MMD, KL-divergence and JS-divergence. In this work, we use the $\epsilon$-*KL-divergence* (Eqn. (2)), whose definition and properties can be found in Appendix B.

**Data batch.** If $|S_t| = 1$, then this is the conventional online learning setting where samples arrive one by one. However, industry practitioners seldom update the model over a single sample, and always collect a batch of samples before fine-tuning the model, so we consider data batches instead.

**Feedback.** Without any feedback from the evaluation process, the problem is fully unsupervised because we only have the unlabeled batches to fine-tune the model. This setting, however, is too hard and unrealistic. Indeed, for most deployed industrial systems, there are tools for evaluating online performance, either through some automated metrics or feedback provided by end-users. Here, we consider the *user error report model*, where a fraction of the users provide feedback on incorrect model outputs. This model leads to *Random Label Feedback (RLF)*, where the labels of $\alpha$ fraction of the samples in $S_t$ are provided as feedback. In this scenario, $\alpha = 0$, $\alpha = 1$ and $\alpha \in (0, 1)$ correspond to the unsupervised, supervised and semi-supervised learning settings, respectively.

**All previous information.** We allow the learner to store all previously seen samples and feedback (though the learner cannot really replay all samples as it would be too inefficient), which is starkly different from most previous papers in CL that assume a limited storage size.

**Recent knowledge.** OCL requires no forgetting on recent knowledge, because (i) in general, practitioners expect the model not to forget too quickly, and (ii) in many applications the same distribution repeats periodically, which makes recent knowledge useful. For example, satellite images in summer and winter look very different (*e.g.* due to snow), but the images in two consecutive summers look similar, so in this case it is useful to remember the knowledge for at least one year.

**Regression set.** In the software industry, *regression* refers to the deterioration of performance after an update (Yan et al., 2021). The *regression set* contains the *regression data* on which making a mistake is more expensive. Moreover, the labeling function $P(Y|X)$ of the regression data changes very little over time if any (no concept shift). The two most common types of regression data in industry are: (i) Frequent data, which appears more often than other data; (ii) Critical data, which weighs more in the model evaluation and whose definition depends on the specific application.

## 2.3 RELATED WORK AND COMPARISON WITH PREVIOUS SETTINGS

This work is related to three areas: Domain adaptation (DA), continual learning (CL), and semi-supervised learning (SSL). DA provides a learner with labeled samples from a source distribution $P$ and (partially labeled, unlabeled, or no) samples from a target distribution $Q$, and requires it to learn a good model on $Q$. Surveys on DA include Lu et al. (2018); Wang & Deng (2018); Ramponi & Plank (2020); Wang et al. (2022b). In CL, the learner needs to continually learn new knowledge from an online stream of data, and settings include task-incremental CL (including domain-incremental and class-incremental CL), task-aware CL, task-agnostic CL, task-free CL and OCL. Surveys on CL include De Lange et al. (2021); Masana et al. (2020); Biesialska et al. (2020). SSL requires the model to learn from a training set consisting of few labeled samples and many unlabeled samples. Surveys on SSL include Van Engelen & Hoos (2020); Ouali et al. (2020); Yang et al. (2021). A more thorough literature review can be found in Appendix A.

There are several differences between the OCL-PDS problem and conventional DA and CL settings. We now explain why our formulation is more relevant and thus useful for industry practitioners.

**PDS vs DA/DG.** Domain adaptation (DA) and its sibling domain generalization (DG) mostly study big, one-shot distribution shifts, *i.e.* training and testing on two very different domains. This has two problems: (i) Practitioners seldom directly apply a model to a very different domain in industry. Instead, they usually train one model for each domain, and train a domain classifier to distinguish among different domains; (ii) Even if a model needs to be applied to a different domain, practitioners would usually first collect some labeled data from the new domain and then fine-tune the model. It is very rare to have no labels or samples at all from the target domain in industry.

On the contrary, OCL-PDS is a very common scenario in industry. First, PDS has been widely reported to cause performance drop in industrial applications (Martinel et al., 2016; Jaidka et al., 2018; Huang & Paul, 2019), and practitioners do not often train new models for PDS. Second, OCL-PDS studies the semi-supervised setting where the practitioners can collect a few labeled samples and many unlabeled samples, which is more realistic than having no labels or samples at all.

**OCL vs CL.** OCL falls under the *task-free continual learning* setting where there is a fixed task and no clear task boundary. It is different from conventional *task-incremental* CL in three ways:

1. Task-incremental CL has $N$ distinct tasks and requires a single model to learn them all, but what practitioners would usually do in this case is to train $N$ models, one for each task. In contrast, in OCL-PDS the task is fixed but the data distribution is gradually and constantly changing, so it is more reasonable to use and continually fine-tune a single model.

2. Conventional CL requires the model to remember *all $N$* tasks with a storage of size $M$. For studying PDS, this requirement has three problems:

    (i) It is not so practical as not all old knowledge is important - A satellite image classifier in 2022 does not need to do very well on images in 2002 with different landscapes.

    (ii) In the "lifelong" setting with $N$ continually growing, we need $M$ to also grow with $N$ (like $M = O(N)$). With a fixed $M$, it is infeasible to remember *everything*.

    (iii) Many applications have *concept shift* where $P(Y|X)$ could change, so remembering old knowledge can be harmful to the performance on the current data distribution. For instance, languages that were not considered offensive 20 years ago are widely recognized as offensive today thanks to the recent civil rights movements.

    Thus, OCL-PDS only requires remembering recent knowledge and important knowledge.

3. Our setting assumes infinite storage unlike previous settings. This is an over-optimistic assumption as there are real applications where the amount of data is so huge that it is impossible to store all data even for big companies. Real applications also have other considerations such as privacy restrictions so that the data cannot be stored forever. However, storage size is rarely the bottleneck of industrial applications. The point of making this assumption is to **not put too much effort into utilizing every bit of storage**. Instead, we want to focus on more practically relevant questions, such as how to leverage unlabeled data and how to improve training efficiency.

**Other similar settings.** First, OCL is different from the task-free CL formulated in some previous work (Aljundi et al., 2019b; de Masson d'Autume et al., 2019; Wang et al., 2022d) where the training is online but the evaluation is offline. Previous work typically splits the data domain into different sections which the learner sees sequentially online, and in the end the learner is evaluated on the entire domain offline. On the contrary, **both training and evaluation in OCL are online**: For each new batch, the model is first tested and then trained on it. Thus, it is possible to have the real "lifelong" learning setting in OCL where the time horizon $T = \infty$ but not in the previous setting.

Second, OCL-PDS is closely related to reinforcement learning (RL) and time series analysis. The difference from RL is that in RL, the agent can learn from a number of episodes, while in OCL-PDS the evaluation is online and one-pass. The difference from time series analysis is that time series focuses on predicting on future data and does not care about forgetting. Moreover, though PDS naturally resides in time series data, in our literature review (Appendix A) we find little work about handling PDS with time series analysis. One such line of work is temporal covariate shift (TCS) (Du et al., 2021) that assumes that $P(Y \mid X)$ is always fixed, which is not assumed in OCL-PDS.

Third, there are two related settings, *gradual domain adaptation (GDA)* (Kumar et al., 2020) and *gradual concept drift* (Liu et al., 2017; Xu & Wang, 2017), that also study gradual shift from one

domain to another with a series of distributions $P_0, P_1, \cdots, P_T$, where $P_0$ is the source domain, $P_T$ is the target domain, and $P_t$ and $P_{t+1}$ is close for each $t$. Both settings only require good adaptation performance and do not consider forgetting. However, **the concept of regression set widely exists in modern deep learning applications**, and no forgetting on the regression set is a critical issue.

Finally, Cai et al. (2021) proposed a similar OCL setting, where the model is also first evaluated on the new batch and then fine-tuned on it. However, OCL-PDS has three important distinctions: (i) Cai et al. (2021) considered a fully supervised setting while OCL-PDS mainly uses a semi-supervised setting that is more common in practice; (ii) OCL-PDS uses different evaluation metrics, and in particular measures the critical regression set performance; (iii) Cai et al. (2021) evaluates knowledge retention only at three time steps whereas OCL-PDS is purely online and evaluates all metrics at all time steps. A more detailed comparison can be found in Appendix A.2.1.

## 3 BENCHMARKS

In search of benchmarks for this novel OCL-PDS problem setup, we first investigate the benchmarks used in previous work on continual learning and find that most of them are one of the following: (i) Multiple datasets, *e.g.* Task 1 is MNIST (LeCun & Cortes, 2010), Task 2 is SVHN (Netzer et al., 2011) and so on; (ii) Perturbed image classification, where different tasks are constructed by rotating the images with different angles, shifting the colors with different scales or permuting the pixels with different random seeds; (iii) Split-class classification, *e.g.* split the 100 classes in CIFAR-100 (Krizhevsky et al., 2009) into 20 groups with 5 classes per group, and make each group a 5-way classification task. None of them is realistic enough to represent the real PDS in practice.

There are two existing PDS benchmarks: CLOC (Cai et al., 2021) and CLEAR (Lin et al., 2021). Both are image classification tasks and do not have regression sets. Thus, we build a new, more comprehensive suite of benchmarks that covers both language and vision tasks, and both classification and regression tasks, with intuitively defined regression sets. Since it is impossible for us to cover all existing tasks, we also provide our 3-step procedure to build our benchmarks, which can be used to construct PDS benchmarks on other existing datasets.

### 3.1 THE 3-STEP PROCEDURE TO BENCHMARK OCL-PDS

Here we provide the 3-step procedure we use to benchmark OCL-PDS on an existing dataset:

1. Separate the data into groups (domains). For example, group the data by year. Then, do an **OOD check** which verifies that there is a significant distribution shift across the groups.

2. Assign shifting group weights to the batches to create a distribution shift across the groups.[1] Then, do a **shift continuity check** which verifies that the shift is continuous (not abrupt).

3. Design a separate regression set which does not intersect with any online batch. Then, do a **regression check** which verifies that naïve methods have regression on this set.

Moreover, for each batch (including the regression set), we randomly divide the batch into a training batch and a test batch. The initial training set contains both the first batch and the training regression set. The learner sees the training batches during OCL, and the separate test batches are used to evaluate the recent and regression set performances. The model is never evaluated on training samples it has already seen because it is not useful as the learner can store all these samples in its buffer.

**Example: CivilComments-WPDS.** Here we briefly demonstrate this procedure and a detailed description can be found in Appendix C.1. The CivilComments dataset contains online comments with topic labels, and we want to model PDS with shifting hot topics on it. In Step 1, we divide the comments into four groups according to their topics, and verify that a model trained on any three groups does poorly on the fourth group; In Step 2, we construct a weight shift among the groups to simulate PDS, and verify that a model trained on labeled samples from previous distributions can do well on the new distribution, so that the shift is continuous; In Step 3, we define the regression set, and verify that catastrophic forgetting will happen if we only train on the new data.

---

[1]This is not equivalent to *group shift* or *subpopulation shift* as studied in fair machine learning and long-tailed learning (learning with imbalanced classes). The group weights here are used to control the scale of the divergence from being too big. We simulate a gradual shift by continuously shifting the group weights.

Table 1: Benchmarks we build. $T + 1 =$ total number of batches. $w =$ recent time window.

| Benchmark | Description | Regression Set | $T+1$ | $w$ |
|---|---|---|---|---|
| CivilComments-WPDS | Toxic language detection on social media | Critical data | 16 | 5 |
| FMoW-WPDS | Satellite image classification for facilities | Frequent data | 25 | 6 |
| Amazon-WPDS | Review sentiment analysis on Amazon.com | Frequent data | 18 | 6 |
| Poverty-WPDS | Satellite image regression for wealth index | Critical data | 14 | 5 |

### 3.2 Our Benchmarks

We build 4 new benchmarks following the guidelines in Appendix C. See Table 1 for a summary. All 4 benchmarks are based on the Wilds datasets (Koh et al., 2021). Please refer to this paper for the potential leverage, broader context and ethic considerations of these datasets. Here we briefly describe the 4 benchmarks we release in this work, and details can be found in Appendix C.

**CivilComments-WPDS.** This benchmark is based on the CivilComments-Wilds dataset (Borkan et al., 2019), which is a toxic language detection task on social media (Figure 1b). WPDS stands for Wilds-PDS. This benchmark models the shift in hot topics. The regression set contains critical data - Comments with severe harassment, including identity attack and explicit sexual comments.

**FMoW-WPDS.** This benchmark is based on the FMoW-Wilds dataset (Christie et al., 2018), which is a satellite image facility classification task (Figure 1a). It models the shift in time. The regression set contains frequent data - Data from two highly populated regions: Americas and Asia.

**Amazon-WPDS.** This benchmark is based on the Amazon-Wilds dataset (Ni et al., 2019), which is a review sentiment analysis task on shopping websites. This benchmark models the shift in language use. The regression set contains frequent data - Data from 10 popular product categories.

**Poverty-WPDS.** This benchmark is based on the PovertyMap-Wilds dataset (Yeh et al., 2020), which is a satellite image wealth index *regression task*. This benchmark models the shift in time. The regression set contains critical data - Images from urban areas. Following Koh et al. (2021), performances here are measured by the Pearson correlation between outputs and ground truths.

## 4 OCL Algorithms

An OCL algorithm consists of the following three components:

- **Continual fine-tuning:** How to fine-tune the model on the new data?
- **Knowledge retention:** How to prevent forgetting?
- **Semi-supervised learning:** How to leverage the unlabeled data?

In particular, an OCL algorithm is called *supervised* if it does not have the semi-supervised learning component, and called *semi-supervised* if it does. In the rest of this section we will provide an overview of the OCL algorithms we implement: first baselines, then supervised, and finally semi-supervised. Implementation details of these algorithms can be found in Appendix D.

### 4.1 Naïve Baselines

The naïve baselines are used to measure the difficulty of an OCL-PDS task for interpreting the performances of OCL algorithms. First we have **First Batch Only (FBO)**, where we only train an initial model on the first batch $S_0$ (including the training regression set) with empirical risk minimization (ERM) and use that model till the end, which serves as a lower bound of the online performance as well as an upper bound of the regression set performance (because it directly trains the model on regression data without any forgetting). Then we have **i.i.d. offline**, where for each $t$ we train a model on a separate training set *i.i.d.* sampled from $D_t$ and test it on $S_t$, which serves an approximate upper bound of the online performance that reflects the generalization gap. Finally, we have **New Batch Only (NBO)**, where we train the model on the new data alone and never care about forgetting, which serves as a lower bound of the knowledge retention performance.

### 4.2 Supervised OCL Algorithms

**Rehearsal based methods.** Rehearsal was first introduced in Ratcliff (1990); Robins (1995) to prevent catastrophic forgetting in CL, where historical data is stored in a memory buffer and replayed to the learner. The simplest method is **ER-FIFO** (also called ring buffer (Chaudhry et al., 2019b)),

where ER stands for *experience replay*. There are three sources of data the model needs to learn or remember: new data, recent data and regression data. Thus, ER-FIFO simply fine-tunes the model over the union of these three sets of data. The buffer looks like a first-in-first-out (FIFO) queue as the new batch replaces the oldest one (while the regression set is never removed).

There are some variants of ER-FIFO with different strategies of selecting replay samples. In **ER-FIFO-RW** where RW stands for reweighting, the three data sources are balanced so that they have the same probability of being sampled in stochastic gradient descent (SGD), which is useful when different batches have different sizes (the weights can also be customized to adjust between learning and remembering). In **Maximally Interfered Retrieval (MIR)** (Aljundi et al., 2019a), the model is first "virtually" updated on the new data only, and those previous samples on which the loss increases the most before and after the virtual update are selected. The model is then recovered and fine-tuned with a real update on the selected samples together with the new samples. Similarly, in **MaxLoss** (Lin et al., 2022) there is also a virtual update step, and previous samples with the highest loss after virtual update are selected to be replayed during real update.

**GEM-PDS.** This method is a combination of Gradient Episodic Memory (GEM) (Lopez-Paz & Ranzato, 2017) and Average GEM (A-GEM) (Chaudhry et al., 2019a), and we design it specially for OCL-PDS. Denote the gradients of the loss function on the new data, recent data and regression data by $g_0$, $g_1$ and $g_2$, respectively. Gradient descent along $g_0$ might cause the model to forget recent and important knowledge, so instead we find a "pseudo gradient" $g$ that is close to $g_0$, and gradient descent along $g$ won't lead to forgetting. We solve the following convex optimization problem:

$$\underset{g}{\text{minimize}} \, \|g - g_0\|_2^2 \qquad \text{s.t.} \qquad \langle g, g_1 \rangle \geq 0, \langle g, g_2 \rangle \geq 0 \qquad (1)$$

This problem is always feasible, and the optimal $g^*$ can be found with a simple procedure described in Appendix D Eqn. (3). The constraints ensure that descent along $g^*$ won't increase the loss on the recent and regression data, which can be shown with the Taylor expansion of the loss function.

**Regularization based methods.** The high-level idea of regularization is to keep the model weights close to the initial model which has a good regression set performance, so as to reduce forgetting. Denote the vectorized model weights at time $t$ by $\theta_t$. In **Online L2 Regularization (L2Reg)**, we add a penalty term $\frac{\lambda}{2}\|\theta_t - \theta_0\|_2^2$ to the loss function. In a variant called **Elastic Weight Consolidation (EWC)** (Kirkpatrick et al., 2017), the penalty term is $\frac{\lambda}{2}\|\text{diag}(F_t)(\theta_t - \theta_0)\|_2^2$, where $F_t$ is the Fisher information matrix (FIM) and $\text{diag}(F_t)$ only contains the elements on the diagonal of $F_t$. The FIM ensures that weights that are more influential to the model output change less.

### 4.3 Semi-supervised OCL Algorithms

**Pseudo Labeling (PL).** PL was proposed in Lee (2013) and is also known as *self-training*. Since $D_t$ is close to $D_{t-1}$, if $f_{t-1}$ can do well on $D_{t-1}$, then it is very likely that it can also do well on $D_t$. Based on this observation, for an unlabeled sample $x$ in $S_t$, define its pseudo label simply as $f_{t-1}(x)$. Then, we fine-tune the model $f_{t-1} \to f_t$ on the union of the labeled and the pseudo-labeled sets with any supervised OCL algorithm. We denote a PL method by adding suffix "PL" after this supervised algorithm, such as ER-FIFO-PL. Note that in classification, $f_{t-1}(x)$ is the "hard" label, so training $f_{t-1}$ on this label minimizes its entropy on $x$. Kumar et al. (2020); Wang et al. (2022a) proved the effectiveness of PL in the GDA context. Moreover, pseudo-labeled samples are not replayed for knowledge retention as they could have large label noise.

Furthermore, in our implementation of PL we introduce an innovative *virtual update* step which we find very important in our experiments. The whole algorithm goes as follows:

1. Virtual update: Fine-tune the model $f_{t-1} \to f'_{t-1}$ with ERM on the labeled samples only.

2. Pseudo label the unlabeled samples with the updated model: $x \to (x, f'_{t-1}(x))$.

3. Revert the model weights from $f'_{t-1}$ back to $f_{t-1}$, and fine-tune $f_{t-1} \to f_t$ with any supervised OCL algorithm on the union of labeled and pseudo-labeled samples.

**FixMatch (FM).** FixMatch (Sohn et al., 2020) is a variant of PL. In FM, there are two types of data augmentation: a strong one and a weak one, and the pseudo labels are generated on the weakly augmented samples while the model is fine-tuned on the strongly augmented samples, which leads to a *consistency regularization* effect that makes the model have consistent outputs on the weakly and strongly augmented samples. Currently FixMatch is only implemented for vision tasks. We denote FixMatch by adding suffix "FM" after a supervised algorithm, such as ER-FIFO-FM.

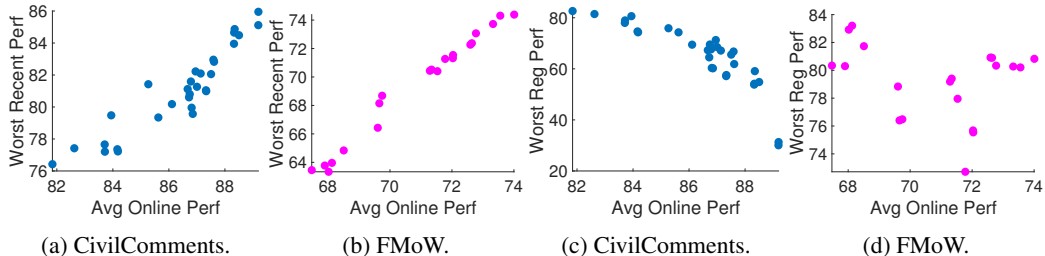

(a) CivilComments.     (b) FMoW.     (c) CivilComments.     (d) FMoW.

Figure 2: Results of supervised OCL algorithms on CivilComments-WPDS ($\alpha = 0.5\%$) and FMoW-WPDS ($\alpha = 50\%$). Each point corresponds to one pair of algorithm and hyperparameters.

## 5 EXPERIMENTS

We compare the algorithms on the 4 benchmarks we constructed for OCL-PDS. Each experiment is run 5 times with different random seeds. For the recent performance and the regression set performance (reg performance), we report both the average and the worst performances, which are the mean and minimum of the performance over $t = 1, \cdots, T$, respectively. The reason why we also report the worst performance is that knowledge retention is required for all $t$. For saving space, we put detailed settings and results in Appendix E. In this section we summarize five remarkable observations we make from our experiments, first supervised and then semi-supervised OCL algorithms.

### 5.1 RESULTS OF SUPERVISED OCL ALGORITHMS

**Observation #1: Strong positive correlation between online and recent performances.** On all benchmarks, methods that achieve higher online performances also achieve higher recent performances. In Figures 2a and 2b, we plot the average online and worst recent performances achieved by different supervised OCL algorithms on the CivilComments-WPDS ($\alpha = 0.5\%$) and FMoW-WPDS ($\alpha = 50\%$) benchmarks ($\alpha$ is the fraction of labeled samples), where we can see a strong positive correlation between these two. The reason is that $D_t$ is very close to $D_{t-1}, \cdots, D_{t-w}$ by formulation, so a model that performs well on $D_t$ naturally performs well on the $w$ recent distributions too. This is also known as the *accuracy-on-the-line* phenomenon (Miller et al., 2021).

**Observation #2: Correlation between online and reg performances differs among benchmarks.** In Figure 2c and 2d, we plot the average online and worst reg performances on CivilComments-WPDS and FMoW-WPDS. We can see that the trends on these two benchmarks are quite different. On CivilComments-WPDS, methods with higher online performances have lower reg performances, while there is no such trend on FMoW-WPDS. One possible reason is that it depends on how the regression set is defined, and how close it is between the regression set distribution and the overall distribution. For CivilComments-WPDS, the regression set samples (severely offensive comments) are very different from the other samples (mostly normal comments), while for FMoW-WPDS the regression set samples from Americas and Asia are closer to the samples from other regions.

**Observation #3: Some existing methods do not work well on regression tasks.** On the right, we plot the average online and worst reg performances of different supervised OCL algorithms on the Poverty-WPDS benchmark ($\alpha = 50\%$), which is a regression task. We observe that while some methods like ER-FIFO and L2Reg do better than the FBO baseline, some others including MIR and EWC achieve lower online performance than FBO which should be a lower bound. Most existing methods have only been tested on classification tasks before, and this experiment shows that they might not work so well on regression tasks.

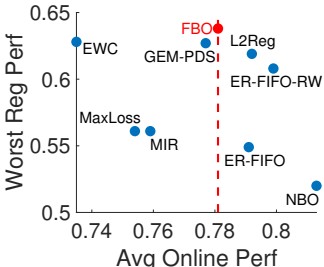

Figure 3: Poverty-WPDS.

### 5.2 RESULTS OF SEMI-SUPERVISED OCL ALGORITHMS

**Observation #4: Unlabeled data improves OCL performance.** In Figure 4, we plot the performances of ER-FIFO-PL/FM and ER-FIFO-RW-PL/FM (in red) along with the performances of all supervised OCL algorithms (in blue) on CivilComments-WPDS and FMoW-WPDS. We can see that with the same worst reg performance, SSL methods achieve higher average online performances, and vice versa. SSL improves the online performance by learning more new data, but it does not help knowledge retention. Thus, supervised OCL algorithms with high online performances (such as MIR) do not improve much with SSL. We also observe that FixMatch is slightly better than PL.

**Observation #5: Virtual update in PL is important.** In Figure 5 we plot the performances of ER-FIFO-PL and ER-FIFO-RW-PL on CivilComments-WPDS with different epochs of virtual update

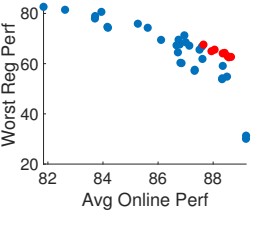

(a) CivilComments.

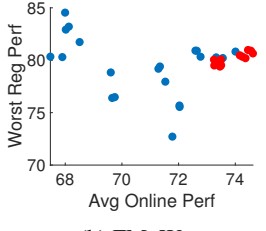

(b) FMoW.

Figure 4: Performances of ER-FIFO-PL/FM and ER-FIFO-RW-PL/FM (in red). FM is only used for FMoW.

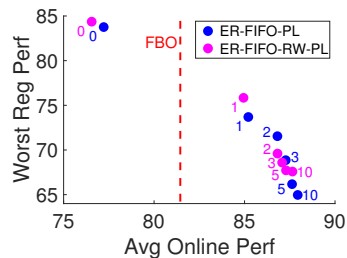

Figure 5: PL with different epochs of virtual update (CivilComments).

as labeled near the points. We can see that without virtual update (0 epoch), the online performance of PL is even lower than the FBO baseline. However, with just 1 epoch of virtual update, the online performance rises above FBO, and with more epochs of virtual update the online performance is higher (but with a lower reg performance). This shows the importance of virtual update though it was not included in previous methods like gradual self-training (Kumar et al., 2020).

One explanation is that the virtual update step distills the knowledge of $P(Y \mid X)$ from the new distribution into the current model, so $P(f'_{t-1}(X) \mid X)$ is closer to $P(Y \mid X, D_t)$. Without virtual update, $f_{t-1}$ only has knowledge from old distributions, so when the model is trained on the pseudo-labeled samples, it reinforces its old knowledge but learns little new knowledge, resulting in a low online performance but high reg performance. This also implies that unsupervised OCL is difficult and perhaps infeasible, because without new labels the learner cannot know how $P(Y \mid X)$ changes.

## 6 DISCUSSION

The application of deep learning has become so wide nowadays that it is very difficult to cover all the tasks with a single general problem formulation. Here we briefly discuss two additional problems that stem from OCL-PDS which practitioners in certain areas in industry might find useful.

**Fine-grained PDS.** In our problem formulation in Section 2.1, the learner can fine-tune the model at every $t$. However, in applications where new data comes in very fast and the distribution changes very quickly, this could be impractical. One typical example is time series analysis such as stock price prediction. We term this setting fine-grained PDS as the data batches are much smaller and are received much more frequently. In our benchmarks, the time horizon $T$ is around 20, while in a fine-grained PDS benchmark $T$ shoud typically be 2,000 or 20,000.

One way to handle fine-grained PDS is to have two fine-tune procedures: A fast one that can be done at every $t$, and a slow one that is run simultaneously with the fast one. For instance, in an ensemble method like Mixture-of-Experts (Masoudnia & Ebrahimpour, 2014), the fast procedure only adjusts the weights of the base models, while the slow procedure trains a new base model with the new data.

**Abrupt shift detection.** We assumed that $\mathrm{Div}(D_t \parallel D_{t+1}) < \rho$ for all $t$. However, this might not always be true, and abrupt shifts could happen in practice. For example, for toxic language detection on social media (Figure 1b), when the US Supreme Court overturned Roe v. Wade, posts related to gender, religion, politics and civil rights flooded on social media, causing a sharp spike in the data distribution and probably a significant drop in the model's performance. Thus, practitioners need a mechanism to detect such abrupt shifts so that prompt human intervention could take place.

Some applications in industry have *online metrics* that keep track of the online performance, and a significant performance drop triggers human intervention. However, in applications without online metrics or where feedback can be delayed, we need to detect distribution shift with unlabeled data alone, a problem known as *OOD detection* (Hendrycks & Gimpel, 2017; Lee et al., 2018). A more difficult recently proposed problem is *OOD performance prediction* (Chen et al., 2021; Jiang et al., 2022; Garg et al., 2022; Baek et al., 2022), *i.e.* predicting the model's performance on the new distribution with unlabeled data alone, as distribution shift does not necessarily hurt the performance.

**Limitation.** While we commit ourselves to studying the real PDS that exists in industrial applications, in this paper for privacy reasons we are limited to working on public datasets, which (despite our effort of making them realistic) are still different from real applications. We hope that in the future there could be more datasets containing real PDS in industry applications released for the better development of this field.

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

# Appendix

## Table of Contents

## A   LITERATURE REVIEW

### A.1   DISTRIBUTION SHIFT

Distribution shift in machine learning refers to the scenario where the model is tested on a distribution $Q$ different from the distribution $P$ on which it was trained, which is different from the conventional machine learning setting where the training and testing sets are *i.i.d.* sampled from the same distribution. Distribution shift is studied in a number of areas in machine learning, including domain adaptation, continual learning, transfer learning, fair machine learning, long-tailed learning, etc. See Table 2 of Gulrajani & Lopez-Paz (2021) and Table 2 of Wang et al. (2022b) for a comparison among these areas.

In general, there are two types of distribution shift problem: Domain shift and subpopulation shift (Koh et al., 2021; Sagawa et al., 2022). In domain shift, the training and testing distributions contain different domains, and the goal is to generalize to new domains. This problem is also called Out-of-distribution (OOD) generalization, such that the training set is ID (in-distribution) and the test set is OOD. Related areas include domain adaptation, domain generalization, transfer learning, etc. In subpopulation shift, the training and testing distributions consist of the same domains, but the relative proportions are different. Related areas include fair machine learning, long-tailed learning (learning with imbalanced classes), etc. The difference between these two is that in subpopulation shift, the supports of the training and testing data distributions are the same, while it is not for domain shift.

A lot of methods have been proposed to train models that are robust to distribution shift. Here we introduce two general methods, and in the subsequent sections we will talk about methods for specific areas. The most classic method is *importance weighting* (Shimodaira, 2000), which multiplies the loss on sample $x$ with an importance weight $\frac{Q(x)}{P(x)}$, because $\int f(x)dQ = \int \frac{Q(x)}{P(x)}dP$. This

method requires that $P(x) > 0$ for all $x$ such that $Q(x) > 0$, *i.e.* only works for subpopulation shift. Another general method that is widely used today is *Distributionally Robust Optimization (DRO)* (Duchi & Namkoong, 2018), which assumes that $Q \in U(P)$ where $U(P)$ is the *uncertainty set* that contains a family of distributions that are close to $P$. Then, DRO trains the model on the worst distribution in $U(P)$ (with the highest empirical risk), so as to ensure that the model can do well on any distribution in $U(P)$ which includes $Q$. A lot of variants of DRO have been proposed and being used today (Hashimoto et al., 2018; Hu et al., 2018; Sagawa et al., 2020a; Lahoti et al., 2020; Zhai et al., 2021a;b). However, there is also a recent line of work that points out some problems with these methods both empirically and theoretically (Byrd & Lipton, 2019; Sagawa et al., 2020b; Gulrajani & Lopez-Paz, 2021; Xu et al., 2021; Wang et al., 2022c). Remarkably, a recent paper Zhai et al. (2022) proved the surprising result that generalized reweighting (GRW) methods, a broad family of methods including importance weighting, DRO and a lot more, cannot do better than ERM for linear models and NTK neural networks (Jacot et al., 2018). Thus, there is still a huge room of improvement for methods for distribution shift.

## A.2 Continual Learning

Continual learning (CL), also known as lifelong learning, comes from the philosophy that a learning agent should be able to continually learn new knowledge and improve itself with new data on its own. As detailed in Ring (1998), a continual learner should be able to learn context-dependent knowledge autonomously, incrementally and hierarchically. A more recent paper Liu (2020) introduced the concept of *on-the-job learning*, where the learner is required to detect new tasks on its own, collect data for the new tasks and then learn with the data. Given these general philosophical ideals, the goal of CL is to mathematically formulate a problem setting that reflects these ideals. Here we point our readers to the Avalanche library (Lomonaco et al., 2021), a recently released Python library that contains many benchmarks and algorithms of continual learning.

### A.2.1 Settings

There are a bunch of continual learning settings, and here we discuss the most widely studied ones.

**Task-incremental CL.** In this setting, $N$ tasks $T_1, \cdots, T_N$ are sequentially given to the learner, who is required to learn these tasks one by one without forgetting the previous tasks. The learner cannot see the old tasks while learning a new one, but it has a memory buffer in which it can store data from previous tasks. The goal is to perform well on all $N$ tasks, and the performance is usually evaluated with the average or the minimum of the performances on all tasks. A related setting is class-incremental CL where new classes sequentially appear. Task-incremental CL is the oldest setting that dates back to McCloskey & Cohen (1989), which trained a feed-forward network to first learn addition with one and then learn addition with two, and found that when learning the second task the network forgot the first one. This work pinpointed the *catastrophic forgetting* problem, and as a result many researchers today believe that "the central problem of continual learning is to overcome the catastrophic forgetting problem" (Aljundi et al., 2019c).

Two related problems are domain-incremental CL and class-incremental CL. In domain-incremental CL, the data of each task comes from a different domain. In class-incremental CL, samples from new classes appear one by one.

**Task-aware CL.** In this setting, the tasks are not sequentially given. Rather, each sample has a "task descriptor" indicating which task it belongs to. This setting dates back to the early paper Ratcliff (1990) which trained a multi-layer encoder model to learn four vectors A, B, C and D, which were provided in a cyclic fashion: ABCDABCDABCD... Then, Lopez-Paz & Ranzato (2017) studied task-aware CL in the online learning setting where samples come in an online stream. The goal of task-aware CL is the same as task-incremental CL: To perform well on all $N$ tasks.

**Task-agnostic CL.** In task-agnostic CL, each sample still belongs to a certain task, but the "task descriptor" is not provided. The model is still required to do well on all $N$ tasks, and it is still evaluated by the average or the minimum of the performances on all tasks. Algorithms for this setting usually do *task inference*, i.e. inferring the task from the input (Van de Ven & Tolias, 2019), where they assume that samples from the same task are closer to each other. Note that our definition

is different from some previous work, and some previous work on task-agnostic CL like Zeno et al. (2018) is in fact under the task-free setting.

**Task-free CL.**  In task-free CL, there is no task at all. The data comes in an online stream and each sample only appears once. This setting was first introduced in Aljundi et al. (2019b), where they split a data domain into several partitions, and sequentially present each partition to the model. At the end of training, the model is evaluated on the entire data domain, *i.e.* the union of all the partitions. Thus, if the learner is allowed to store all samples and train the model on them, then the problem becomes equivalent to supervised learning. However, the model cannot store all samples because the storage size is assumed to be limited. This problem is also called the *data incremental* learning problem in De Lange et al. (2021). We can see that in this problem, the training is online (partitions are sequentially given), but the evaluation is offline (test once on the entire domain).

**Online Continual Learning (OCL).**  We study OCL in this work. It is a variant of task-free CL as there is no task in OCL. The key characteristic of OCL is that *both training and evaluation are online*, which is different from the previous task-free CL problem. In OCL, for each new data batch, the model is first evaluated on it and then trained on it. Thanks to the online evaluation, we can study the real "lifelong" learning setting in OCL where the time horizon is infinite, which is not possible in the previous task-free CL problem. Note that the term "online continual learning" was used in quite a few previous papers, but most of them refer to "continual learning with online data" (such as Yin et al. (2021a)) which is not the OCL setting we define in this work.

**Time series analysis.**  A related area is time series analysis, where the data comes from a non-stationary online distribution, and the task is to predict future data with the current and past observations. We can see that distribution shift naturally exists in the formulation of time series analysis, but we fail to find much work about dealing with distribution shift with time series analysis.

One setting in time series analysis that is similar to OCL is temporal covariate shift (TCS), which was first introduced in Du et al. (2021). TCS makes the covariate shift assumption: $P(Y|X)$ is fixed while $P(X)$ shifts with time. The goal of TCS is $r$-step ahead prediction, *i.e.* predicting the labels for inputs whose ground truth labels will be revealed $r$ steps later. Other recent papers that study distribution shift with time series analysis include Duan et al. (2022); Gagnon-Audet et al. (2022); Kim et al. (2022).

**Comparison with a previous work.**  One previous work Cai et al. (2021) proposed an OCL framework similar to ours. In both settings, the model is evaluated on new online batches before fine-tuned on them. The differences between our work and Cai et al. (2021) are the following:

1. Cai et al. (2021) considered a fully supervised setting: The environment reveals all true labels to the learner after evaluation. In contrast, in OCL-PDS, we use the random label feedback which randomly selects $\alpha$ fraction of the new samples and provides their labels. In particular, we mainly study a semi-supervised learning setting where the environment only reveals a fraction of the labels, which is a more common scenario in industrial applications.

2. The evaluation metrics used in Cai et al. (2021) were: Average online accuracy, backward transfer and forward transfer. The first two metrics correspond to the average online performance and the average recent knowledge retention in OCL-PDS. In addition to these metrics, OCL-PDS also considers the regression set performance which is a very important metric in real applications, the worst (recent/important) knowledge retention performances since knowledge retention is required for all t, and the training efficiency which is very important in an online setting where the algorithm is run for many times.

3. Cai et al. (2021) only evaluates knowledge retention performances at three fixed time steps: $H/3$, $2H/3$ and $H$, where $H$ is the total number of time steps. In contrast, OCL-PDS is *purely online*: all metrics including knowledge retention are evaluated at each step.

4. Cai et al. (2021) proposed the CLOC benchmark which is indeed a PDS benchmark. However, CLOC only covers vision classification, while our benchmarks cover both language and vision tasks, and both classification and regression tasks.

5. Cai et al. (2021) only run experience replay (ER) on their benchmark, while our work also compares regularization based methods such as EWC, variants of ER such as GEM,

MIR and MaxLoss, as well as semi-supervised learning methods like pseudo-labeling and FixMatch.

### A.2.2 METHODS

**Rehearsal based methods.** The concept of "rehearsal" was first introduced in Ratcliff (1990); Robins (1995). Rehearsal-based methods store samples in an operational memory (a memory buffer). When the learner learns a new task, it also reviews the old samples in this buffer in order to prevent catastrophic forgetting, known as *experience replay*. Almost all previous work assumed that the buffer has a fixed size, so two key questions for these methods are: (i) Which samples to be store in the buffer, and (ii) Which samples to be replayed to the learner.

For problem (i), the most widely used strategy is Reservoir sampling (Isele & Cosgun, 2018) which maintains the buffer distribution to be the average of all task distributions. Other variants include Chrysakis & Moens (2020); Kim et al. (2020). For problem (ii), MIR (Aljundi et al., 2019a) selects the samples with the highest loss increase after a "virtual update" step to be replayed, MaxLoss (Lin et al., 2022) selects the samples with the highest loss after virtual update, and OCS (Yoon et al., 2022) selects an online coreset that is diverse and has a high affinity to previous tasks. Moreover, experience replay (ER) can be combined with other methods. For example, MER (Riemer et al., 2019) combines ER with meta learning, GMED (Jin et al., 2021) combines ER with adversarial attack, and Wang et al. (2022d) combines ER with DRO.

There are alternative ways to use this buffer. For example, iCaRL (Rebuffi et al., 2017) stores "exemplars" in the buffer and uses a nearest neighbor classifier with these exemplars. In other words, samples in the buffer are not used for training, but used for inference. Similarly, Continual Prototype Evolution (De Lange & Tuytelaars, 2021) maintains and continually updates a prototype for each class, and uses a nearest neighbor classifier for inference. Another way is introduced in GEM (Lopez-Paz & Ranzato, 2017), where the buffered samples are not directly replayed for training, but instead used to find a "pseudo gradient" with a convex optimization problem so that the loss on previous tasks won't increase. Variants of GEM include A-GEM (Chaudhry et al., 2019a) and GSS (Aljundi et al., 2019c).

Since the buffer size is assumed to be limited, a bunch of previous papers studied how to utilize the buffer space more efficiently. One line of work proposes to learn the distributions of previous tasks with a generative model, which includes DGR (Shin et al., 2017), FearNet (Kemker & Kanan, 2018) and so on.

**Regularization based methods.** The high-level idea of these methods is to not change the model weights too much so that the model's performance on previous tasks won't drop too much. One type of methods directly add to the objective a penalty term which keeps the model weights close to the old weights, such as EWC (Kirkpatrick et al., 2017). Another type of methods use *synaptic regularization* (Zenke et al., 2017; Aljundi et al., 2019b) which controls the learning rate of each weight, so that more influential weights change more slowly.

**Architecture based methods.** These methods continually update the model architecture in order to learn the new tasks. One type of method changes the architecture by adding a "mask" to some weights, such as Piggyback (Mallya et al., 2018) which learns a mask for each task, learning a hard attention mask with gradient descent (Serra et al., 2018), and NCCL that adds weight calibration modules and feature calibration modules to balance between stability and plasticity (Yin et al., 2021b). Another type of method uses the isolation approach where the model has two parts - A shared part and a task specific part. When a new task arrives, the shared part is updated very little, and a new task specific part is learned on the new task. Examples of isolation methods include progressive neural network (PNN) (Rusu et al., 2016), Learning without Forgetting (LwF) (Li & Hoiem, 2017), dynamically expandable network (DEN) (Yoon et al., 2018), etc.

**Semi-supervised/Unsupervised continual learning.** There are also some existing continual learning methods that work under a semi-supervised or unsupervised setting. For instance, Lump (Madaan et al., 2022) uses Mixup to interpolate between the current task and previous tasks' instances to alleviate catastrophic forgetting, and Fini et al. (2022) combines continual learning with self-supervised learning.

### A.3 DOMAIN ADAPTATION

Domain adaptation (DA) is a type of distribution shift where "the tasks are the same, and the differences are only caused by domain divergence" (Wang & Deng, 2018). There is a source distribution $P$ and a target distribution $Q$ with distribution shift from $P$ to $Q$. During training, the learner is provided with samples from both the source and the target distributions, and conditioning on whether the samples from the target distribution is labeled, partially labeled or unlabeled, DA is classified as supervised, semi-supervised and unsupervised DA. Particularly, in supervised DA, the number of target samples is usually very small so that training on these target samples alone cannot lead to a good model. A related area is *domain generalization* (Gulrajani & Lopez-Paz, 2021; Blanchard et al., 2021) where the learner does not have any samples from the target domain, even unlabeled ones.

In general, a DA algorithm consists of two parts: Feature alignment and class alignment. The goal of *feature alignment* is to train a feature encoder $\Phi$ that can encode *invariant features*, *i.e.* the images of the source and target domains in the feature space are close to each other, or $\Phi(P) \approx \Phi(Q)$. As summarized in Wang & Deng (2018), there are two common ways to achieve feature alignment. The first one is adversarial-based, *i.e.* training a domain discriminator to distinguish features from the source and target domains, and the features are aligned if this discriminator cannot achieve a high performance. Methods of this type include DANN (Ganin et al., 2016), SagNet (Nam et al., 2021), etc. The second one is discrepancy-based, *i.e.* minimizing a divergence function between $\Phi(P)$ and $\Phi(Q)$, also called the "confusion alignment loss" in Motiian et al. (2017). Methods of this type include CORAL (Sun & Saenko, 2016), IRM (Arjovsky et al., 2019), ARM (Zhang et al., 2021), etc. However, there are also some papers that point out the problems within these methods (Rosenfeld et al., 2021; Gulrajani & Lopez-Paz, 2021).

Moreover, even if we have learned a feature encoder $\Phi$ such that $\Phi(P) \approx \Phi(Q)$, we cannot be sure that the same classifier works for both domains, because samples of different classes in $P$ and $Q$ might be mapped to the same latent feature. Thus, the goal of *class alignment* is either to make sure that samples of the same class are mapped together, or to train a new classifier $w'$ which works for $\Phi(Q)$. Note that class alignment requires labels from the target domain which are unavailable in unsupervised domain adaptation or domain generalization. For instance, Tzeng et al. (2015) used soft labels for class alignment, Long et al. (2016) minimized the cross entropy on the target data while using a residual block to keep the source and target classifiers close, and Motiian et al. (2017) used a similarity penalty between samples from different classes.

### A.4 SEMI-SUPERVISED LEARNING

In many applications, the number of labeled samples are limited, but there are also a large number of unlabeled samples. For example, for image classification tasks, while the labels are hard to obtain, free images can be very easily retrieved from the internet. Semi-supervised learning studies how to train a model on a small set of labeled samples and a large set of unlabeled samples. This is an old area of research and there is a very rich body of work, on which we do not intend to make an exhaustive survey here. Here we briefly discuss several types of methods that are widely being used today, and more methods can be found in surveys such as Van Engelen & Hoos (2020).

**Generating labels for unlabeled samples.** Perhaps the most direct and intuitive way of leveraging the unlabeled samples is to try to generate labels for them, and then train the model over all samples as if they were all labeled. The simplest of such methods is pseudo-labeling (Lee, 2013), which first trains a model over the labeled samples alone and then uses this model to pseudo-label the unlabeled samples. This method assumes that the labeled and unlabeled samples come from the same underlying distribution, so if a model can do well on the labeled samples, it should be able to generate good pseudo labels for the unlabeled samples. However, the quality of pseudo labels depends on the generalization ability of the model, and if the number of labeled samples is too small, then the pseudo labels could contain large systematic label noise. Thus, a number of techniques have been proposed to improve pseudo-labeling.

One such technique is *consistency regularization*, which is based on the following observation: Given two transformations $x'$ and $x''$ of the same sample $x$, their labels should be the same. For instance, FixMatch (Sohn et al., 2020) uses two augmentation methods: a weak one and a strong

one. It generates pseudo labels on the weakly augmented sample $x'$, and trains the model on the strongly augmented one $x''$. Similarly, Noisy Student (Xie et al., 2020) also uses a weak and a strong augmentation, but it alternates between teacher phases which generate pseudo labels and student phases which learn these labels, until convergence. Some other work defines $x'$ and $x''$ as the outputs of the model at different epochs, such as Temporal Ensembling (Laine & Aila, 2017) and Mean Teachers (Tarvainen & Valpola, 2017). There is also a line of work that leverages adversarial attack, such as Virtual Adversarial Training (VAT) (Miyato et al., 2018).

**Interpolation based methods.** These methods train the model on the interpolation between labeled and unlabeled samples. This type of methods was initially introduced in MixUp (Zhang et al., 2018), which interpolates between two labeled samples to combat label noise and adversarial attack as it makes the model have linear behavior in between samples. This method is then applied to semi-supervised learning in MixMatch (Berthelot et al., 2019), which combines MixUp with a lot of other techniques including pseudo labeling.

**Self-supervised learning.** The high-level idea of self-supervised learning (also known as representation learning) is to train a good feature extractor on the unlabeled data set with an auxiliary task (upstream task), and then train a classifier on top of it on the labeled data set (downstream task). The most famous and widely-used self-supervised learning technique is masked language modeling (MLM) in NLP (Devlin et al., 2018), where the auxiliary task is predicting a masked word within a sentence. MLM has achieved great success in NLP as the feature extractor it learns can be applied to almost any language task and lead to good performance.

Inspired by the success of MLM, people also try to apply self-supervised learning to vision tasks. For example, Doersch et al. (2015) extracts random pairs of patches from each image where the auxiliary task is to learn the relative position between each pair of patches, and Gidaris et al. (2018) rotates each image with different angles where the auxiliary task is to learn this rotation angle, which is applied to semi-supervised learning in $S^4L$ (Zhai et al., 2019). Today, the most widely used technique is *contrastive learning*, which extracts different views from each image, and the auxiliary task is to learn which views come from the same image. The feature extractor is trained to learn the similarity between two views: similar if they come from the same image, and different if they do not. This idea was first introduced in Bachman et al. (2019). Currently the most popular contrastive learning methods include SimCLR (Chen et al., 2020), MoCo (He et al., 2020), BYOL (Grill et al., 2020), SwAV (Caron et al., 2020), etc.

## B  THE DIVERGENCE FUNCTION

In this section, we dive deep into one important problem in the formulation of the OCL-PDS problem: How to choose the divergence function $\text{Div}(P \parallel Q)$ that guarantees the continuity of the distribution shift? As mentioned in Section 2.2, $\text{Div}(P \parallel Q)$ refers to the divergence *from $P$ to $Q$*, and ideally we want it to reflect *the performance of a model which is trained on $P$ and tested on $Q$*.

To study this problem, first we will review existing divergence functions, and then we will show that an ideal divergence function for OCL-PDS should be *asymmetric and bounded*, which is unfortunately not satisfied by any popular divergence function. Finally, we will introduce the *$\epsilon$-KL-divergence* which is used in this work.

### B.1  EXISTING DIVERGENCE FUNCTIONS

This part is based on the NeurIPS tutorial by Gretton et al. (2019). Generally speaking, there are two types of existing divergence functions: *Integral probability metrics (IPMs)* and *$\phi$-divergences*. To quickly understand these two types of divergence, think about how to determine whether two distributions $P$ and $Q$ are equal. There are two ways in general: (a) Compute $P - Q$ and see if it is zero almost everywhere, and (b) Compute $P/Q$ and see if it is one almost everywhere. IPMs correspond to method (a) and $\phi$-divergences correspond to method (b).

**IPMs.** An IPM is defined as $\text{Div}(P \parallel Q) = \sup_{f \in \mathcal{F}} \left[ \mathbb{E}_{X \sim P} f(X) - \mathbb{E}_{Y \sim Q} f(Y) \right]$ for some function family $\mathcal{F}$. Examples include total variation (TV) defined as $\text{Div}(P \parallel Q) = \frac{1}{2} \int |P(x) -$

$Q(x)|dx$, MMD defined as $\text{Div}(P \parallel Q) = \|\mathbb{E}_{X \sim P}[\pi(X)] - \mathbb{E}_{Y \sim Q}[\pi(Y)]\|_{\mathcal{H}}$ for some feature mapping $\pi$ and reproducing kernel Hilbert space $\mathcal{H}$, and the Wasserstein distance defined as $\text{Div}(P \parallel Q) = \inf_{\gamma \in \Gamma(P,Q)} \int D(x,y) d\gamma(x,y)$ for some distance function $D(\cdot, \cdot)$.

$\phi$**-divergences.** A $\phi$-divergence is defined as $\text{Div}_\phi(P \parallel Q) = \int \phi\left(\frac{dP}{dQ}\right) dQ$. For example, when $\phi = -\log$, then the $\phi$-divergence becomes the reverse KL-divergence, where the popular KL-divergence is defined as $D_{KL}(Q \parallel P) = \int_x Q(x) \log\left(\frac{Q(x)}{P(x)}\right) dx$ (note that $P$ and $Q$ are reversed). Total variation is the only non-trivial function that is both an IPM and a $\phi$-divergence.

## B.2 Divergence Function for OCL-PDS

In this part, we show that an ideal divergence function for the OCL-PDS problem should be asymmetric and bounded.

**Asymmetric.** Suppose we have two very different data domains $A$ and $B$. Let $P = A$ and $Q = 0.5A + 0.5B$. A model trained on $P$ would have a very poor performance on $Q$, because it has never seen any samples from domain $B$. On the other hand, a model trained on $Q$ could have a good performance on $P$, because it has seen samples from both $A$ and $B$. Thus, in this example, we would like to have $\text{Div}(P \parallel Q) > \text{Div}(Q \parallel P)$, so Div should be asymmetric.

**Bounded.** The KL-divergence is widely used in machine learning literature, but one problem is that it is unbounded. Recall that in the OCL-PDS problem formulation, we assume that $\text{Div}(D_t \parallel D_{t+1}) < \rho$ for all $t$. Now consider what would happen if the function Div is unbounded, and we want to introduce data from new domains into the problem. Specifically, $Q$ contains samples from new domains that are not in $P$, *i.e.* there exists $x$ such that $Q(x) > 0$ and $P(x) = 0$. In this situation, we must have $\text{Div}(P \parallel Q) = \infty$, no matter how small $Q(x)$ is. Therefore, **if Div is unbounded like the reverse KL-divergence, then we could never introduce new domains into the problem,** which is not desirable. There is a variant of the KL-divergence called JS-divergence, which is defined as $\text{Div}(P \parallel Q) = \frac{1}{2}[D_{KL}(P \parallel M) + D_{KL}(Q \parallel M)]$ where $M = \frac{1}{2}P + \frac{1}{2}Q$. Although it is bounded, it is also symmetric, so it is not ideal for OCL-PDS.

## B.3 $\epsilon$-KL-Divergence

In this work, we use the $\epsilon$-*KL-divergence* defined as follows:

$$\text{Div}(P \parallel Q) = \mathbb{E}_{X \sim P}[g(X)] + \mathbb{E}_{Y \sim Q}[\log(-g(Y))] + 1$$
$$\text{where} \quad g(x) = -\frac{Q(x)}{\max\{P(x), \epsilon\}} \tag{2}$$

This divergence functions has the following properties:

(i) This function is a lower bound of the reverse KL-divergence, and it is bounded.

(ii) If $g(x) = -\frac{Q(x)}{P(x)}$, then this function is equivalent to the reverse KL-divergence (which is its dual formulation). Thus, if for any $x$ such that $Q(x) > 0$, we have $P(x) \geq \epsilon$, then the $\epsilon$-KL-divergence is equivalent to the reverse KL-divergence.

(iii) In the case where $Q$ contains new domains that are not in $P$, for example $Q = (1-\beta)P + \beta\tilde{P}$ for some $\tilde{P} \perp P$, then $\text{Div}(P \parallel Q) = \beta + \beta\log\frac{\beta}{\epsilon} + (1-\beta)\log(1-\beta) \approx \beta + D_{KL}(Q \parallel (1-\epsilon)P + \epsilon\tilde{P})$.

From the properties, we can see that what we are doing in the $\epsilon$-KL-divergence is essentially adding an $\epsilon$ lower bound to the denominator of $g(x)$ so that the function becomes bounded.

**Example.** Given two groups $A$ and $B$, let $\rho = 0.17$ and $\epsilon = 0.02$. Then, we have the following group weight allocation schedule which is widely used in our benchmarks:

Table 2: Sample group weight allocation schedule.

| $t$ | $A$ | $B$ | $\text{Div}(D_t \parallel D_{t+1})$ |
|---|---|---|---|
| 0 | 1.00 | 0.00 | 0.1661 |
| 1 | 0.90 | 0.10 | 0.1674 |
| 2 | 0.69 | 0.31 | 0.1663 |
| 3 | 0.41 | 0.59 | 0.1595 |
| 4 | 0.15 | 0.85 | 0.1625 |
| 5 | 0.00 | 1.00 | |

**Limitations.** The major limitation of the $\epsilon$-KL-divergence is that it cannot measure how similar two domains are. For example, suppose we have three domains: $A$, $B$ and $C$, and they do not overlap with one another. Samples in $A$ and $B$ are very similar, but they are vastly different from samples in $C$. In this case, a model trained on $A$ can have a good performance on $0.5A + 0.5B$, but a poor performance on $0.5A + 0.5C$. However, for the $\epsilon$-KL-divergence, we have $\text{Div}(A \parallel 0.5A + 0.5B) = \text{Div}(A \parallel 0.5A + 0.5C)$. We can see that the $\epsilon$-KL-divergence cannot measure the similarity between $A$ and $B$ or $A$ and $C$. Another limitation is that the choice of $\epsilon$ is arbitrary and can affect the function value. Nevertheless, even with these two limitations, we still believe that the $\epsilon$-KL-divergence is suitable for the OCL-PDS problem. Finally, keep in mind that no divergence function can cover every facet of real problems in practice, and that's why we have three important checking steps in our benchmarking procedure described in Section 3.1.

## C  BENCHMARK DETAILS

We build 4 new benchmarks for OCL-PDS following the guidelines listed below:

- We only use public datasets in this work.
- The benchmarks should cover a wide variety of tasks.
- The benchmarks should be realistic and can reflect the real PDS in industry.
- Naïve methods should be poor on these benchmarks, so special methods are necessary.

In this section, we will first demonstrate in detail how to use the 3-step procedure described in Section 3.1 with the CivilComments-WPDS benchmark as an example. Then, we will present the details for all other datasets, but without the detailed benchmarking procedure.

### C.1  CIVILCOMMENTS-WPDS

Here we present how we construct CivilComments-WPDS from the CivilComments-Wilds dataset with the 3-step procedure including the 3 important checking steps.

**Step 1: Separate the data into groups.** First, we investigate the metadata we have in this dataset. In CivilComments-Wilds, apart from the target label, each sample also has the following annotations: whether it contains a certain topic (male, female, LGBTQ, christian, muslim, other religions, black, white), and whether it contains a certain type of toxicity (identity attack, explicit sexual, etc.). Based on this metadata, we can model the distribution shift with the shift in hot topics. We separate the samples into four groups according to their topics, as shown in the following table.

Table 3: Samples in CivilComments-Wilds are separated into 4 groups by their topics.

| Group | Topic | # Samples | # Toxic | # Non-toxic |
|---|---|---|---|---|
| 0 | Normal | 269,422 | 21,297 | 248,125 |
| 1 | Race | 37,459 | 10,820 | 26,639 |
| 2 | Religion | 65,816 | 8,106 | 57,710 |
| 3 | Gender | 75,303 | 10,571 | 64,732 |

We can observe that: (i) There are less toxicity in normal comments than comments about a specific topic; (ii) The portion of toxic comments about race is much higher than that of other topics.

Then, we do the **OOD check**, where we verify that there is a distribution shift across the groups. We perform this check in the following way: For each Group $k$, we train a model on a training set sampled from all groups except Group $k$, and then test this model on a validation set sampled from all groups except Group $k$, and a test sampled from Group $k$. The results are the following:

Table 4: OOD check results of CivilComments-WPDS (%).

| Group $k$ | Topic | Validation Accuracy | Test Accuracy | OOD Gap: Val - Test |
|---|---|---|---|---|
| 0 | Normal | 87.14 | 94.52 | -7.38 |
| 1 | Race | 92.90 | 78.86 | 14.04 |
| 2 | Religion | 92.11 | 89.88 | 2.23 |
| 3 | Gender | 92.20 | 89.80 | 2.40 |

From this table, we can see that Group 1 (race) has the largest OOD performance gap, much larger than other groups. Thus, when allocating the groups in Step 2, we will make sure that the shift to Group 1 is slower. Moreover, we observe that on Group 0, the gap is negative, which means that the OOD performance is better than the ID performance (*i.e.* A model trained on Groups 1-3 has a higher accuracy on Group 0 than Groups 1-3). This is a counter-intuitive phenomenon as it is usually taken for granted that OOD performance should be lower than ID performance. The cause of this phenomenon might be that Group 0 is much easier to learn than the other groups (for example, the data is more concentrated and depends on fewer features).

**Step 2: Assign shifting group weights to the batches.** We want to model the distribution shift with the shift in hot topics on social media. At each time $t$, $D_t$ contains normal comments as well as comments about the current hot topic, *i.e.* samples from Group 0 exists in every batch. We design the schedule listed in the following table:

Table 5: Group weight schedule for CivilComments-WPDS. For each $t$, the weights add up to 1.

| $t$ | Group 0 | Group 1 | Group 2 | Group 3 |
|---|---|---|---|---|
| 0 | 1.00 | 0.00 | 0.00 | 0.00 |
| 1 | 0.90 | 0.10 | 0.00 | 0.00 |
| 2 | 0.75 | 0.25 | 0.00 | 0.00 |
| 3 | 0.60 | 0.40 | 0.00 | 0.00 |
| 4 | 0.40 | 0.60 | 0.00 | 0.00 |
| 5 | 0.40 | 0.30 | 0.30 | 0.00 |
| 6 | 0.40 | 0.00 | 0.60 | 0.00 |
| 7 | 0.40 | 0.00 | 0.30 | 0.30 |
| 8 | 0.40 | 0.00 | 0.00 | 0.60 |

Then, we do the **shift continuity check**, whose point is to make sure that the distribution shift in this schedule is continuous. This check goes as follows: For each $t$, we sample a set from $D_t$ and split it into a training set and a validation set, and then sample a test set from $D_{t+1}$. We want to make sure that the gap between the validation and the test performances is not too large for each $t$. The results are the following:

Table 6: Shift continuity check results of CivilComments-WPDS (%).

| $t$ | Validation Accuracy | Test Accuracy | OOD Gap: Val - Test |
|---|---|---|---|
| 0 | 94.87 | 92.99 | 1.88 |
| 1 | 93.49 | 91.31 | 2.18 |
| 2 | 90.67 | 88.10 | 2.57 |
| 3 | 87.56 | 83.59 | 3.97 |
| 4 | 85.60 | 89.18 | -3.58 |
| 5 | 89.08 | 91.59 | -2.51 |
| 6 | 91.63 | 91.69 | -0.06 |
| 7 | 92.05 | 91.47 | 0.58 |

We can see that the OOD gaps are kept under 4%, as opposed to the 14.04% gap we got in the OOD check. And we can observe the same phenomenon again: For some $t$, the OOD accuracy is higher than the ID occuracy.

Then, based on these results, we design the following sample allocation schedule listed in Table 8. The first batch contains 50000 samples, while all the other batches contain 10000 samples each.

**Step 3: Design a separate regression set.** We notice that each toxic comment is annotate which types of toxicity the comment contains in the CivilComments-Wilds dataset. Thus, we design the regression set as the set of comments with two types of severe toxicity: identity attack and explicit sexual. Such comments are critical toxic comments that a good detector should be able to detect with high success rate.

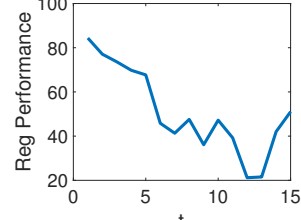

Figure 6: Regression check.

Then, we do the **regression check**, where we verify that naïve methods have regression on the regression set we constructed. Here we use the NBO baseline described in Section 4.1 as the "naïve method". As shown in Figure 6, the regression set performance of NBO quickly declines to under 50% - The performance of random guessing. Thus, the benchmarks passes this check.

**Train/test split.** For each $t$, we split $S_t$ into a training set and a test set, and the regression set is also split into a training set and a test set. These test sets are used to evaluate the recent and regression set performances. We never evaluate the learner on data it has seen, because the learner has an infinitely large buffer where it can store the data. Instead, we evaluate the learner on separate *i.i.d.* test sets. Note that the numbers listed in Table 8 are all sizes of the training sets.

## C.2 FMOW-WPDS

This benchmark is built from FMoW-Wilds. First, we separate the data into 5 groups by the year:

Table 7: Samples in FMoW-WPDS are clustered into 5 groups by their year.

| Group | Year | # Samples |
|---|---|---|
| 0 | 2002-2013 | 132,948 |
| 1 | 2014 | 54,575 |
| 2 | 2015 | 87,358 |
| 3 | 2016 | 140,459 |
| 4 | 2017 | 54,746 |

Table 8: Sample allocation schedule for CivilComments-WPDS.

| $t$ | Group 0 | Group 1 | Group 2 | Group 3 |
|---|---|---|---|---|
| 0 | 50000 | 0 | 0 | 0 |
| 1 | 9000 | 1000 | 0 | 0 |
| 2 | 9000 | 1000 | 0 | 0 |
| 3 | 7500 | 2500 | 0 | 0 |
| 4 | 7500 | 2500 | 0 | 0 |
| 5 | 6000 | 4000 | 0 | 0 |
| 6 | 6000 | 4000 | 0 | 0 |
| 7 | 4000 | 6000 | 0 | 0 |
| 8 | 4000 | 3000 | 3000 | 0 |
| 9 | 4000 | 3000 | 3000 | 0 |
| 10 | 4000 | 0 | 6000 | 0 |
| 11 | 4000 | 0 | 6000 | 0 |
| 12 | 4000 | 0 | 3000 | 3000 |
| 13 | 4000 | 0 | 3000 | 3000 |
| 14 | 4000 | 0 | 0 | 6000 |
| 15 | 4000 | 0 | 0 | 6000 |
| Total | 131000 | 27000 | 24000 | 18000 |

Table 9: Sample allocation schedule for FMoW-WPDS.

| $t$ | Group 0 | Group 1 | Group 2 | Group 3 | Group 4 |
|---|---|---|---|---|---|
| 0 | 120000 | 0 | 0 | 0 | 0 |
| 1 | 0 | 10000 | 0 | 0 | 0 |
| 2 | 0 | 9000 | 1000 | 0 | 0 |
| 3 | 0 | 6900 | 3100 | 0 | 0 |
| 4 | 0 | 6900 | 3100 | 0 | 0 |
| 5 | 0 | 4100 | 5900 | 0 | 0 |
| 6 | 0 | 4100 | 5900 | 0 | 0 |
| 7 | 0 | 1500 | 8500 | 0 | 0 |
| 8 | 0 | 0 | 10000 | 0 | 0 |
| 9 | 0 | 0 | 9000 | 1000 | 0 |
| 10 | 0 | 0 | 6900 | 3100 | 0 |
| 11 | 0 | 0 | 6900 | 3100 | 0 |
| 12 | 0 | 0 | 4100 | 5900 | 0 |
| 13 | 0 | 0 | 4100 | 5900 | 0 |
| 14 | 0 | 0 | 1500 | 8500 | 0 |
| 15 | 0 | 0 | 1500 | 8500 | 0 |
| 16 | 0 | 0 | 0 | 10000 | 0 |
| 17 | 0 | 0 | 0 | 9000 | 1000 |
| 18 | 0 | 0 | 0 | 6900 | 3100 |
| 19 | 0 | 0 | 0 | 6900 | 3100 |
| 20 | 0 | 0 | 0 | 4100 | 5900 |
| 21 | 0 | 0 | 0 | 4100 | 5900 |
| 22 | 0 | 0 | 0 | 1500 | 8500 |
| 23 | 0 | 0 | 0 | 1500 | 8500 |
| 24 | 0 | 0 | 0 | 0 | 10000 |
| Total | 120000 | 42500 | 71500 | 80000 | 46000 |

Then, we allocate the groups with the schedule listed in Table 9. We put 120000 samples from Group 0 into batch 0 for pretraining, and all other batches contain 10000 samples each. The regression set is defined as the set of samples in two highly populated regions: Americas and Asia.

### C.3 AMAZON-WPDS

This benchmark is built from the Amazon-Wilds dataset. In the original paper (Koh et al., 2021), the ID/OOD sets are divided by the reviewers, and the dataset also contains some other metadata such as year and product categories. However, we find in our experiments that these groups cannot create a sufficiently large distribution shift. Thus, we use the following class split method, which has been widely used in the continual learning literature: We divide the 5 classes of this datasets (corresponding to the 5 stars rating) into 2 groups and 2 classes - positive and negative reviews, as shown in the following table:

Table 10: In Amazon-WPDS, the 5 stars rating are divided into 2 groups and 2 classes.

|  | Group 0 | Group 1 | # Samples | Group 0 | Group 1 |
|---|---|---|---|---|---|
| **y = 0** | 1, 2 stars | 3 stars | **y = 0** | 193,900 | 377,039 |
| **y = 1** | 4 stars | 5 stars | **y = 1** | 1,087,385 | 2,343,846 |

Then, we allocate the groups to the batches with the schedule listed in Table 12. This schedule is based on the weight schedule we obtained in Table 2. The first batch has 50000 samples from Group 1, and then we model a group shift from Group 1 to Group 0. Each subsequent batch has 5000 samples.

Finally, the regression set is defined as the set of reviews from 10 popular product categories, including books, fashion, etc.

### C.4 POVERTY-WPDS

This benchmark is build from the PovertyMap-Wilds dataset, which is a image regression task. First, we cluster the data into 4 group by the year:

Table 11: Samples in Poverty-WPDS are clustered into 4 groups by their year.

| Group | Year | # Samples |
|---|---|---|
| 0 | 2009-2011 | 7129 |
| 1 | 2012-2013 | 5005 |
| 2 | 2014 | 3494 |
| 3 | 2015-2016 | 4041 |

Then we allocate the groups with the schedule listed in Table 13. We put all samples from Group 0 (except test samples) into the first batch for pretraining, and each subsequent batch contains 800 samples. Finally, the regression set is defined as the samples from urban areas, which can better reflect the overall wealth index of each country. Note that this is a regression task, and we evaluate the model performance with the Pearson correlation following Koh et al. (2021).

## D ALGORITHM DETAILS

First of all, at $t = 0$, all OCL algorithms train the initial model on the first labeled batch and the training regression set with empirical risk minimization (ERM). In particular, following Koh et al. (2021), we use the cross entropy loss for classification tasks and the mean squared error for regression tasks. The differences among the methods only appear after $t > 0$.

**Baselines.** First, note that the upper bound baseline **i.i.d. offline** is not an OCL algorithm because it assumes access to $D_t$ at time $t$. For each $t$, it trains a model on a set sampled from $D_t$ with a sufficiently large size and no overlapping with $S_t$. The size of the training set is different for different benchmarks, but we always make sure that it is at least as large as the union of all batches before time $t$. **i.i.d. offline** is only an approximate upper bound of the online performance because

Table 12: Sample allocation schedule for Amazon-WPDS.

| $t$ | Group 0 | Group 1 |
|---|---|---|
| 0 | 0 | 50000 |
| 1 | 500 | 4500 |
| 2 | 500 | 4500 |
| 3 | 1550 | 3450 |
| 4 | 1550 | 3450 |
| 5 | 1550 | 3450 |
| 6 | 2950 | 2050 |
| 7 | 2950 | 2050 |
| 8 | 2950 | 2050 |
| 9 | 2950 | 2050 |
| 10 | 4250 | 750 |
| 11 | 4250 | 750 |
| 12 | 4250 | 750 |
| 13 | 4250 | 750 |
| 14 | 5000 | 0 |
| 15 | 5000 | 0 |
| 16 | 5000 | 0 |
| 17 | 5000 | 0 |
| Total | 54450 | 80550 |

Table 13: Sample allocation schedule for Poverty-WPDS.

| $t$ | Group 0 | Group 1 | Group 2 | Group 3 |
|---|---|---|---|---|
| 0 | 6359 | 0 | 0 | 0 |
| 1 | 0 | 800 | 0 | 0 |
| 2 | 0 | 720 | 80 | 0 |
| 3 | 0 | 720 | 80 | 0 |
| 4 | 0 | 552 | 248 | 0 |
| 5 | 0 | 552 | 248 | 0 |
| 6 | 0 | 328 | 400 | 72 |
| 7 | 0 | 328 | 400 | 72 |
| 8 | 0 | 120 | 400 | 280 |
| 9 | 0 | 120 | 400 | 280 |
| 10 | 0 | 0 | 400 | 400 |
| 11 | 0 | 0 | 200 | 600 |
| 12 | 0 | 0 | 0 | 800 |
| 13 | 0 | 0 | 0 | 800 |
| Total | 6359 | 4240 | 2856 | 3304 |

for a distribution $P$, training on multiple distributions other than $P$ might even achieve a better performance on $P$ than training on $P$ directly, as we saw in benchmarking CivilComments-WPDS. However, this baseline gives us a sense of the generalization error, and thus help us interpret the performances of other OCL algorithms.

Then, for **FBO**, the algorithm does not do anything after $t > 0$, and the initial model is used till the end. For **NBO**, for each new batch $S_t$, the model is trained on the labeled portion of $S_t$ only with a fixed number of epochs of ERM, and no previous sample is replayed.

**ER-FIFO and ER-FIFO-RW.** In ER-FIFO, for each $t > 0$, the model is fine-tuned on the union of the new labeled batch, the recent labeled batches, and the training regression set, for a fixed number of epochs (named `epochs`) of ERM. Specifically, the recent labeled batches and the training regression set are stored in the memory buffer. One issue of this approach is that it does not take

into account the sizes of the batches and the regression set. For example, if the regression set is much larger than the batches, then it will be hard for the model to learn the new knowledge. Thus, ER-FIFO-RW alters the weights of the three data sets: the new data, the recent data and the regression data. Specifically, it uses uniform sampling over these three sets, so that each set has the same probability of being selected.

**MIR and MaxLoss.**   When there are too many previous samples in the buffer, we cannot replay all of them for efficiency. ER-FIFO-RW solves this problem by randomly sampling previous samples to replay. However, this might not be the most efficient method, as some previous samples might be more useful than the others for preventing catastrophic forgetting. MIR and MaxLoss are two strategies of selecting replay samples, which operate as follows: For each iteration,

1. Sample $n$ new samples and $n_{kr}$ previous samples that require knowledge retention (KR).

2. Virtual update: Fine-tune $f_{t-1} \to f'_{t-1}$ with ERM on the $n$ new samples.

3. Select $n$ samples from the $n_{kr}$ previous samples for replay. Specifically, MIR selects the samples whose loss increase the most before and after virtual update, while MaxLoss selects the samples with the highest loss after virtual update.

4. Recover the model to $f_{t-1}$, and fine-tune $f_{t-1} \to f_t$ with ERM on the $n$ new samples and the $n$ selected previous samples.

The ratio $n_{kr}/n$ is named as `kr_size` in our code, which should be greater than 1. The larger $n_{kr}$ is, the more likely we can select "useful" replay samples, but the slower the algorithm.

**GEM-PDS.**   In this method, for each iteration, we first sample $n$ new data, $n_{kr}$ recent data and $n_{kr}$ regression data, on which we estimate the three gradients of the loss function $g_0$, $g_1$ and $g_2$, respectively. A larger $n_{kr}$ allows the learner to estimate $g_1$ and $g_2$ more accurately. The ratio $n_{kr}/n$ is still named as `kr_size`. Then, the pseudo gradient $g^*$ is the optimal solution of the convex optimization problem Eqn. (1), which can be solved with the following procedure:

1.  $a = \langle g_0, g_1 \rangle, b = \langle g_1, g_1 \rangle, c = \langle g_1, g_2 \rangle, d = \langle g_0, g_2 \rangle, e = \langle g_2, g_2 \rangle$.

2.  If $a \geq 0, d \geq 0$ then return $g_0$.

3.  $p = cd - ae, q = ac - bd, r = be - c^2$.

4.  $\hat{g}_1 = g_0 - \dfrac{a}{b} g_1$. If $a \leq 0$ and $q \leq 0$ then return $\hat{g}_1$.

5.  $\hat{g}_2 = g_0 - \dfrac{d}{e} g_2$. If $d \leq 0$ and $p \leq 0$ then return $\hat{g}_2$.

6.  Return $\hat{g}_3 = g_0 + \dfrac{p}{r} g_1 + \dfrac{q}{r} g_2$.

(3)

We can verify with the KKT conditions that this procedure returns the correct solution of Eqn. (1) (*e.g.* see Section 5.5.3 of Boyd et al. (2004)). The model is then updated with gradient descent along $g^*$.

**Online L2 Regularization and EWC.**   Let the loss of model $f_t$ parameterized by $\theta_t$ (which is the vectorized model weight) on the labeled batch of $S_t$ be $\ell_t(\theta_t)$. In online L2 regularization, we minimize the following objective function with a fixed number of epochs of ERM:

$$\min_{\theta_t} \quad \ell_t(\theta_t) + \frac{\lambda}{2} \|\theta_t - \theta_0\|_2^2 \qquad (4)$$

where we add a $L_2$ penalty between the new model weight $\theta_t$ and the initial model weight $\theta_0$. The reasons why we use $\|\theta_t - \theta_0\|_2^2$ instead of $\|\theta_t - \theta_{t-1}\|_2^2$ are:

1. $\theta_0$ has a very good regression set performance, which is not guaranteed for $\theta_{t-1}$. So using $\|\theta_t - \theta_0\|_2^2$ ensures a higher regression set performance.

2. If $T$ is very large, then the model weight can still change a lot if we use $\|\theta_t - \theta_{t-1}\|_2^2$ (as the change accumulates with $t$), so knowledge retention cannot be guaranteed.

In EWC, the objective function is the following:

$$\min_{\theta_t} \quad \ell_t(\theta_t) + \frac{\lambda}{2}\|\text{diag}(F_t)(\theta_t - \theta_0)\|_2^2 \tag{5}$$

where $F_t$ is the Fisher information matrix (FIM), and $\text{diag}(F_t)$ only contains the elements on the diagonal of $F_t$. Following Kirkpatrick et al. (2017), we estimate $\text{diag}(F_t)$ from the first-order derivatives of the loss function on $n_{kr}$ samples for knowledge retention. The ratio $n_{kr}/n$ is still named as `kr_size`. With a larger $n_{kr}$, we can estimate $F_t$ more accurately. $\lambda$ is named as `lbd`.

**Pseudo Labeling (PL) and FixMatch (FM).** In PL and FM, there are two hyperparameters: the number of epochs of the virtual update step `epochs_v`, and the number of epochs of the real fine-tuning step `epochs_r`. Virtual update is done with ERM on the labeled samples only, while real fine-tuning is done with a supervised OCL algorithm on the union of labeled and pseudo-labeled samples. Thus, PL and FM can be combined with any supervised OCL algorithm, which we denote by adding the suffix "PL" or "FM" to the algorithm, such as ER-FIFO-PL and ER-FIFO-FM.

## E  EXPERIMENT DETAILS

### E.1  SETUP

Following Koh et al. (2021), we use a DistilBert-base-uncased for CivilComments-WPDS and Amazon-WPDS, a DenseNet-121 for FMoW-WPDS, and a ResNet-18 for Poverty-WPDS. For the training hyperparameters, we generally use the same ones as in Koh et al. (2021), with one exception that we use a multi learning rate decay scheduler for the two vision benchmarks as we find that it can produce better performances than the old scheduler.

For each experiment we report the following 6 metrics:

- For online performance, we report the **average online performance (avg online)** within a finite $T$.
- For knowledge retention, we report the **average recent performance (avg recent)** and the **worst recent performance (worst recent)**, which is the average and the minimum of the recent performance over $t = w, \cdots, T$. We also report the worst performance because knowledge retention is required for every $t$. Likewise, we also report the **average reg performance (avg reg)** (regression set performance) and the **worst reg performance (worst reg)**. As mentioned in Section 3.1, we use separate test batches to evaluate the recent and reg performances.
- For training efficiency, we report the **average runtime (avg time)** of the method over $t = 1, \cdots, T$. Note that the time used to train the initial model ($t = 0$) is not computed in the average runtime.

Each experiment is run on a single NVIDIA V100 GPU. Each experiment is run 5 times with different random seeds, and the mean and the standard deviation of the results are reported. In particular, for each benchmark we use 5 fixed initial models: We train 5 initial models on the first batch and the training regression set with ERM for a fixed number of epochs with different random seeds, and then use these 5 models as initial models for all methods. This both alleviates the effect of randomness in the initial models and saves time. Under this setting, FBO can also serve as an upper bound of the worst regression set performance, which is the regression set performance at $t = 1$ for every method.

### E.2  RESULTS

In Table 14 we list the notations we use in the results. The results are reported in Tables 15-18.

Table 14: Notations of algorithms.

| Algorithm | Hyperparameters | Notation | Example |
|---|---|---|---|
| NBO | `epochs` | NBO-`epochs` | NBO-5 |
| ER-FIFO | `epochs` | ER-FIFO-`epochs` | ER-FIFO-5 |
| ER-FIFO-RW | `epochs` | ER-FIFO-RW-`epochs` | ER-FIFO-RW-5 |
| MIR | `epochs`, `kr_size` | MIR-`epochs`-`kr_size` | MIR-5-4 |
| MaxLoss | `epochs`, `kr_size` | MaxLoss-`epochs`-`kr_size` | MaxLoss-5-4 |
| GEM-PDS | `epochs`, `kr_size` | GEM-PDS-`epochs`-`kr_size` | GEM-PDS-5-4 |
| L2Reg | `epochs`, `lbd` | L2Reg-`epochs`-`lbd` | L2Reg-5-0.1 |
| EWC | `epochs`, `lbd`, `kr_size` | L2Reg-`epochs`-`lbd`-`kr_size` | EWC-5-0.1-4 |
| ER-FIFO-PL | `epochs_v`, `epochs_r` | ER-FIFO-PL-`epochs_v`-`epochs_r` | ER-FIFO-PL-10-5 |
| ER-FIFO-RW-PL | `epochs_v`, `epochs_r` | ER-FIFO-RW-PL-`epochs_v`-`epochs_r` | ER-FIFO-RW-PL-10-5 |
| ER-FIFO-FM | `epochs_v`, `epochs_r` | ER-FIFO-FM-`epochs_v`-`epochs_r` | ER-FIFO-FM-10-5 |
| ER-FIFO-RW-FM | `epochs_v`, `epochs_r` | ER-FIFO-RW-FM-`epochs_v`-`epochs_r` | ER-FIFO-RW-FM-10-5 |

Table 15: Results on CivilComments-WPDS ($\alpha = 0.5\%$). Accuracies in %.

| Algorithm | Avg Online | Avg Recent | Worst Recent | Avg Reg | Worst Reg | Avg Time |
|---|---|---|---|---|---|---|
| FBO | 81.45 (0.37) | | | | 84.79 (0.96) | |
| i.i.d. offline | 90.03 (0.19) | | | | | |
| NBO-100 | 89.19 (0.39) | 89.08 (0.23) | 85.12 (1.28) | 55.20 (7.54) | 30.07 (11.67) | 95.61 (0.85) |
| ER-FIFO-1 | 82.62 (0.34) | 82.92 (0.30) | 77.42 (0.97) | 83.81 (0.41) | 81.46 (1.00) | 309.04 (1.12) |
| ER-FIFO-RW-3 | 85.27 (0.70) | 85.87 (0.54) | 81.42 (1.44) | 80.93 (0.57) | 75.90 (1.71) | 7.75 (0.04) |
| ER-FIFO-RW-10 | 83.94 (0.31) | 84.09 (0.47) | 79.48 (0.64) | 83.32 (0.43) | 80.63 (1.09) | 24.48 (0.04) |
| ER-FIFO-RW-100 | 81.84 (0.59) | 81.95 (0.78) | 76.42 (1.54) | 84.48 (0.68) | 82.64 (1.10) | 248.83 (3.52) |
| MIR-10-4 | 87.33 (0.30) | 86.70 (0.51) | 80.99 (1.30) | 75.94 (1.98) | 57.57 (3.95) | 40.99 (0.12) |
| MIR-100-4 | 84.16 (0.15) | 83.36 (0.38) | 77.36 (1.11) | 82.29 (0.97) | 74.65 (0.95) | 419.84 (2.08) |
| MIR-10-10 | 86.85 (0.29) | 86.06 (0.58) | 79.57 (1.42) | 77.71 (1.43) | 60.21 (4.29) | 75.31 (0.37) |
| MIR-100-10 | 83.72 (0.18) | 83.09 (0.28) | 77.21 (0.48) | 83.00 (0.65) | 78.96 (1.60) | 757.66 (5.90) |
| MaxLoss-10-4 | 87.32 (0.31) | 86.76 (0.51) | 81.04 (0.80) | 76.25 (1.90) | 57.15 (2.91) | 31.27 (0.09) |
| MaxLoss-100-4 | 84.18 (0.44) | 83.36 (0.40) | 77.22 (0.79) | 82.39 (1.32) | 74.31 (1.74) | 317.14 (1.47) |
| MaxLoss-10-10 | 86.81 (0.34) | 86.05 (0.46) | 79.96 (1.11) | 77.63 (1.31) | 60.35 (2.31) | 52.46 (0.41) |
| MaxLoss-100-10 | 83.71 (0.18) | 83.12 (0.35) | 77.66 (0.58) | 82.73 (0.95) | 77.99 (1.03) | 502.09 (6.19) |
| GEM-PDS-10-4 | 87.59 (0.39) | 87.64 (0.28) | 82.96 (1.01) | 76.13 (1.38) | 66.78 (4.01) | 68.15 (0.07) |
| GEM-PDS-30-4 | 87.61 (0.55) | 87.58 (0.45) | 82.83 (1.01) | 74.92 (2.53) | 61.88 (8.54) | 195.42 (0.29) |
| GEM-PDS-10-10 | 86.95 (0.43) | 86.86 (0.46) | 82.22 (0.59) | 78.33 (1.37) | 71.28 (2.31) | 149.77 (0.14) |
| GEM-PDS-30-10 | 86.72 (0.43) | 86.60 (0.62) | 80.60 (1.61) | 77.52 (1.71) | 64.50 (6.38) | 456.54 (0.72) |
| L2Reg-10-1.0 | 88.35 (0.38) | 88.51 (0.42) | 84.87 (0.81) | 70.51 (3.19) | 59.10 (3.88) | 10.33 (0.13) |
| L2Reg-10-10.0 | 87.13 (0.48) | 87.20 (0.61) | 82.09 (0.62) | 75.63 (2.90) | 67.15 (4.56) | 9.82 (0.12) |
| L2Reg-10-100.0 | 86.11 (0.35) | 86.03 (0.51) | 80.18 (0.97) | 78.07 (2.07) | 69.46 (5.63) | 9.76 (0.12) |
| L2Reg-100-1.0 | 87.54 (0.39) | 87.35 (0.42) | 82.04 (0.71) | 74.56 (2.21) | 65.30 (3.71) | 94.03 (0.67) |
| L2Reg-100-10.0 | 86.91 (0.37) | 86.65 (0.49) | 80.94 (0.91) | 76.30 (2.21) | 68.32 (2.54) | 91.92 (0.71) |
| L2Reg-100-100.0 | 86.18 (0.35) | 85.93 (0.46) | 79.98 (1.04) | 78.05 (1.87) | 69.80 (3.40) | 93.58 (0.39) |
| EWC-10-1.0-4 | 88.50 (0.45) | 88.54 (0.55) | 84.49 (1.28) | 68.74 (3.76) | 54.83 (5.17) | 38.86 (0.37) |
| EWC-10-10.0-4 | 87.51 (0.25) | 87.39 (0.45) | 82.06 (1.16) | 74.67 (2.09) | 65.57 (2.81) | 39.00 (0.44) |
| EWC-10-100.0-4 | 85.62 (0.18) | 85.38 (0.38) | 79.34 (0.79) | 79.75 (1.46) | 74.30 (3.14) | 38.81 (0.08) |
| EWC-100-1.0-4 | 86.56 (0.32) | 86.36 (0.45) | 80.70 (0.98) | 77.53 (2.04) | 71.48 (3.50) | 371.97 (1.27) |
| EWC-100-10.0-4 | 86.24 (0.31) | 86.01 (0.48) | 80.24 (1.01) | 78.21 (1.80) | 72.01 (3.56) | 371.44 (0.86) |
| EWC-100-100.0-4 | 85.20 (0.21) | 84.79 (0.43) | 78.73 (0.82) | 80.44 (1.44) | 75.44 (3.06) | 368.17 (3.44) |
| ER-FIFO-PL-0-1 | 77.22 (1.83) | 76.54 (2.06) | 69.66 (2.17) | 85.66 (1.31) | 83.76 (0.87) | 510.40 (8.05) |
| ER-FIFO-PL-1-1 | 85.21 (0.57) | 86.46 (0.48) | 82.81 (0.79) | 79.54 (0.72) | 73.69 (2.45) | 526.34 (4.13) |
| ER-FIFO-PL-3-1 | 87.28 (0.33) | 88.22 (0.38) | 84.59 (1.02) | 76.26 (1.49) | 68.86 (2.63) | 514.51 (4.14) |
| ER-FIFO-PL-10-1 | 87.94 (0.35) | 88.76 (0.32) | 85.72 (0.44) | 73.82 (2.14) | 64.97 (4.44) | 501.86 (1.50) |
| ER-FIFO-PL-RW-0-1 | 76.55 (1.05) | 75.92 (1.97) | 69.13 (2.96) | 86.39 (0.76) | 84.36 (1.08) | 499.08 (1.22) |
| ER-FIFO-PL-RW-1-1 | 84.94 (0.85) | 85.97 (0.91) | 82.32 (1.44) | 80.26 (1.40) | 75.84 (1.36) | 531.59 (7.74) |
| ER-FIFO-PL-RW-3-1 | 87.08 (0.30) | 88.29 (0.28) | 84.79 (0.97) | 76.60 (0.94) | 68.59 (0.56) | 522.92 (8.41) |
| ER-FIFO-PL-RW-10-1 | 87.65 (0.46) | 88.64 (0.37) | 85.57 (0.63) | 75.28 (2.13) | 67.58 (3.41) | 512.99 (2.48) |

Table 16: Results on FMoW-WPDS ($\alpha = 50\%$). Accuracies in %.

| Algorithm | Avg Online | Avg Recent | Worst Recent | Avg Reg | Worst Reg | Avg Time |
|---|---|---|---|---|---|---|
| FBO | 68.00 (0.24) | | | | 84.54 (0.21) | |
| i.i.d. offline | 83.04 (0.25) | | | | | |
| NBO-5 | 71.78 (0.08) | 73.03 (0.06) | 71.27 (0.26) | 76.84 (0.23) | 72.71 (0.47) | 161.90 (2.11) |
| ER-FIFO-2 | 74.01 (0.10) | 76.04 (0.12) | 74.38 (0.34) | 83.91 (0.09) | 80.82 (0.49) | 1458.19 (239.67) |
| ER-FIFO-RW-2 | 72.77 (0.04) | 74.56 (0.16) | 73.07 (0.15) | 81.84 (0.17) | 80.34 (0.30) | 226.31 (10.83) |
| ER-FIFO-RW-5 | 73.33 (0.11) | 75.21 (0.06) | 73.72 (0.28) | 82.33 (0.12) | 80.27 (0.34) | 480.61 (20.08) |
| ER-FIFO-RW-10 | 73.56 (0.09) | 75.80 (0.08) | 74.31 (0.29) | 82.63 (0.08) | 80.21 (0.19) | 953.56 (96.59) |
| MIR-2-2 | 69.75 (0.09) | 70.74 (0.25) | 68.68 (0.57) | 79.25 (0.25) | 76.48 (0.66) | 357.74 (3.26) |
| MIR-2-4 | 71.29 (0.06) | 72.36 (0.10) | 70.44 (0.20) | 80.81 (0.16) | 79.19 (0.52) | 460.35 (9.11) |
| MIR-2-10 | 72.59 (0.08) | 73.95 (0.08) | 72.27 (0.15) | 81.90 (0.25) | 80.92 (0.20) | 710.30 (54.07) |
| MaxLoss-2-2 | 69.66 (0.21) | 70.53 (0.30) | 68.15 (0.81) | 79.23 (0.13) | 76.40 (0.56) | 318.70 (3.33) |
| MaxLoss-2-4 | 71.34 (0.07) | 72.46 (0.05) | 70.52 (0.32) | 80.86 (0.12) | 79.40 (0.35) | 361.38 (25.58) |
| MaxLoss-2-10 | 72.64 (0.08) | 74.00 (0.10) | 72.37 (0.21) | 81.90 (0.30) | 80.90 (0.32) | 553.68 (18.74) |
| GEM-PDS-2-2 | 72.07 (0.08) | 73.21 (0.19) | 71.49 (0.33) | 79.10 (0.15) | 75.19 (0.45) | 240.80 (3.06) |
| GEM-PDS-2-4 | 72.04 (0.11) | 73.14 (0.09) | 71.54 (0.34) | 79.16 (0.23) | 75.54 (0.75) | 502.77 (41.03) |
| GEM-PDS-2-10 | 72.05 (0.07) | 73.21 (0.13) | 71.45 (0.37) | 79.17 (0.21) | 75.86 (0.38) | 969.29 (5.54) |
| L2Reg-2-0.1 | 69.61 (0.11) | 70.14 (0.19) | 66.43 (0.28) | 81.59 (0.17) | 78.84 (0.95) | 76.42 (1.15) |
| L2Reg-2-1.0 | 67.90 (0.14) | 68.32 (0.20) | 63.78 (0.61) | 82.30 (0.21) | 80.30 (1.00) | 79.84 (1.27) |
| L2Reg-2-10.0 | 68.02 (0.12) | 68.74 (0.26) | 63.34 (0.43) | 83.74 (0.19) | 82.92 (0.30) | 79.94 (0.99) |
| EWC-2-0.1-4 | 70.16 (0.16) | 70.76 (0.24) | 67.72 (0.43) | 81.94 (0.10) | 80.20 (0.60) | 239.01 (1.32) |
| EWC-2-1.0-4 | 68.51 (0.16) | 69.01 (0.27) | 64.84 (0.49) | 82.93 (0.14) | 81.73 (0.42) | 288.53 (26.63) |
| EWC-2-10.0-4 | 68.13 (0.12) | 68.84 (0.20) | 63.96 (0.35) | 83.69 (0.11) | 83.21 (0.14) | 287.97 (26.06) |
| ER-FIFO-PL-3-3 | 74.25 (0.15) | 76.34 (0.22) | 74.49 (0.48) | 84.00 (0.18) | 80.34 (0.43) | 1881.01 (6.93) |
| ER-FIFO-PL-5-3 | 74.36 (0.08) | 76.47 (0.23) | 75.00 (0.58) | 83.95 (0.13) | 80.20 (0.50) | 1924.03 (15.96) |
| ER-FIFO-PL-3-5 | 74.55 (0.06) | 76.54 (0.24) | 74.96 (0.42) | 84.40 (0.10) | 80.90 (0.60) | 3071.01 (28.59) |
| ER-FIFO-PL-5-5 | 74.62 (0.05) | 76.67 (0.13) | 74.91 (0.21) | 84.29 (0.11) | 80.65 (0.23) | 3128.51 (54.12) |
| ER-FIFO-PL-RW-3-3 | 73.40 (0.04) | 75.28 (0.25) | 73.80 (0.40) | 82.12 (0.16) | 79.80 (0.37) | 573.71 (9.83) |
| ER-FIFO-PL-RW-5-3 | 73.49 (0.09) | 75.43 (0.18) | 73.90 (0.34) | 82.10 (0.19) | 80.07 (0.24) | 622.28 (6.41) |
| ER-FIFO-PL-RW-3-5 | 73.50 (0.15) | 75.41 (0.26) | 73.70 (0.46) | 82.27 (0.08) | 79.52 (0.14) | 891.74 (9.63) |
| ER-FIFO-PL-RW-5-5 | 73.46 (0.04) | 75.42 (0.14) | 73.89 (0.21) | 82.26 (0.06) | 79.40 (0.31) | 944.21 (6.73) |
| ER-FIFO-FM-3-3 | 74.81 (0.09) | 76.91 (0.15) | 75.36 (0.62) | 84.37 (0.11) | 80.80 (0.18) | 2022.45 (179.49) |
| ER-FIFO-FM-5-3 | 74.80 (0.06) | 76.86 (0.13) | 75.27 (0.17) | 84.32 (0.16) | 80.36 (0.42) | 2084.54 (183.02) |
| ER-FIFO-FM-3-5 | 75.10 (0.07) | 77.19 (0.13) | 75.46 (0.15) | 84.70 (0.16) | 80.96 (0.27) | 3056.44 (52.54) |
| ER-FIFO-FM-5-5 | 75.16 (0.05) | 77.33 (0.19) | 75.75 (0.60) | 84.71 (0.10) | 80.88 (0.39) | 3268.89 (307.79) |
| ER-FIFO-RW-FM-3-3 | 73.44 (0.05) | 75.37 (0.30) | 73.80 (0.37) | 81.87 (0.19) | 79.62 (0.35) | 572.95 (2.68) |
| ER-FIFO-RW-FM-5-3 | 73.40 (0.08) | 75.31 (0.12) | 73.83 (0.30) | 81.80 (0.05) | 79.52 (0.41) | 642.91 (37.83) |
| ER-FIFO-RW-FM-3-5 | 73.71 (0.05) | 75.72 (0.15) | 74.04 (0.41) | 82.27 (0.08) | 79.38 (0.44) | 896.77 (4.61) |
| ER-FIFO-RW-FM-5-5 | 73.75 (0.15) | 75.68 (0.13) | 74.32 (0.25) | 82.31 (0.05) | 79.52 (0.33) | 971.52 (39.10) |

Table 17: Results on Amazon-WPDS ($\alpha = 0.5\%$). Accuracies in %.

| Algorithm | Avg Online | Avg Recent | Worst Recent | Avg Reg | Worst Reg | Avg Time |
|---|---|---|---|---|---|---|
| FBO | 84.85 (0.20) | | | | 93.78 (0.26) | |
| i.i.d. offline | 94.15 (0.19) | | | | | |
| NBO-100 | 90.49 (0.41) | 90.68 (0.34) | 88.58 (0.95) | 92.22 (0.43) | 88.63 (0.76) | 105.68 (1.74) |
| ER-FIFO-1 | 87.12 (0.34) | 88.06 (0.28) | 84.70 (0.40) | 93.58 (0.16) | 93.02 (0.28) | 314.64 (1.60) |
| ER-FIFO-RW-10 | 88.47 (0.67) | 89.32 (0.73) | 86.85 (1.62) | 93.63 (0.30) | 93.00 (0.37) | 17.09 (0.06) |
| ER-FIFO-RW-100 | 86.65 (0.21) | 87.39 (0.31) | 83.22 (1.07) | 93.68 (0.20) | 93.11 (0.37) | 181.74 (1.46) |
| MIR-10-4 | 89.46 (0.46) | 90.19 (0.51) | 88.00 (1.55) | 93.31 (0.34) | 91.82 (0.94) | 28.88 (0.23) |
| MIR-100-4 | 86.99 (0.48) | 87.87 (0.59) | 84.28 (0.87) | 93.62 (0.38) | 92.86 (0.67) | 286.59 (2.91) |
| MIR-10-10 | 89.15 (0.61) | 89.95 (0.66) | 87.28 (2.16) | 93.45 (0.36) | 92.21 (0.82) | 47.99 (0.24) |
| MIR-100-10 | 87.16 (0.36) | 88.07 (0.42) | 84.68 (0.38) | 93.78 (0.26) | 93.24 (0.29) | 479.84 (11.69) |
| MaxLoss-10-4 | 89.42 (0.43) | 90.14 (0.54) | 87.52 (2.01) | 93.33 (0.28) | 91.82 (0.84) | 24.37 (0.17) |
| MaxLoss-100-4 | 87.06 (0.69) | 87.88 (0.75) | 84.42 (1.17) | 93.72 (0.33) | 93.13 (0.41) | 247.32 (6.66) |
| MaxLoss-10-10 | 89.32 (0.57) | 90.06 (0.61) | 87.78 (1.78) | 93.49 (0.27) | 92.41 (0.57) | 34.72 (0.24) |
| MaxLoss-100-10 | 87.11 (0.45) | 88.06 (0.66) | 84.73 (1.18) | 93.73 (0.30) | 93.18 (0.30) | 358.98 (3.47) |
| GEM-PDS-10-4 | 89.94 (0.46) | 90.50 (0.41) | 88.74 (1.11) | 92.94 (0.46) | 90.65 (0.47) | 37.30 (0.18) |
| GEM-PDS-30-4 | 90.32 (0.47) | 90.84 (0.35) | 89.22 (1.08) | 92.55 (0.43) | 89.64 (0.66) | 111.27 (0.22) |
| GEM-PDS-10-10 | 89.77 (0.61) | 90.42 (0.51) | 88.20 (1.79) | 93.10 (0.46) | 91.30 (0.45) | 82.24 (0.20) |
| GEM-PDS-30-10 | 90.26 (0.58) | 90.81 (0.55) | 89.32 (1.23) | 92.80 (0.26) | 89.93 (1.16) | 239.08 (0.97) |
| L2Reg-10-0.1 | 89.01 (0.43) | 89.79 (0.45) | 86.72 (2.43) | 93.23 (0.47) | 91.46 (1.00) | 11.06 (0.16) |
| L2Reg-10-1.0 | 88.00 (0.51) | 88.83 (0.57) | 85.14 (1.48) | 93.62 (0.20) | 92.94 (0.35) | 8.52 (0.05) |
| L2Reg-10-10.0 | 86.67 (0.30) | 87.46 (0.35) | 82.21 (1.26) | 93.75 (0.18) | 93.26 (0.18) | 8.53 (0.07) |
| EWC-10-0.1-4 | 89.32 (0.43) | 90.05 (0.46) | 87.06 (2.60) | 93.20 (0.35) | 91.21 (1.21) | 26.12 (0.41) |
| EWC-10-1.0-4 | 88.94 (0.45) | 89.77 (0.49) | 86.48 (2.77) | 93.34 (0.27) | 91.60 (1.27) | 23.02 (0.21) |
| EWC-10-10.0-4 | 88.52 (0.48) | 89.32 (0.57) | 85.75 (2.05) | 93.58 (0.24) | 92.72 (0.34) | 23.34 (0.24) |
| ER-FIFO-PL-0-1 | 85.75 (1.09) | 86.39 (1.27) | 80.91 (2.85) | 93.94 (0.10) | 93.39 (0.17) | 428.25 (11.42) |
| ER-FIFO-PL-1-1 | 87.84 (0.57) | 88.94 (0.69) | 86.21 (1.57) | 93.77 (0.22) | 93.17 (0.27) | 427.75 (7.14) |
| ER-FIFO-PL-3-1 | 88.97 (0.36) | 89.98 (0.26) | 87.72 (0.48) | 93.66 (0.28) | 92.92 (0.42) | 422.17 (2.50) |
| ER-FIFO-PL-10-1 | 89.47 (0.40) | 90.39 (0.38) | 88.42 (1.43) | 93.20 (0.13) | 92.07 (0.36) | 419.94 (1.45) |
| ER-FIFO-RW-PL-0-1 | 85.28 (1.37) | 85.96 (1.41) | 80.30 (2.62) | 93.66 (0.44) | 92.97 (0.58) | 277.10 (0.58) |
| ER-FIFO-RW-PL-1-1 | 88.21 (0.95) | 89.03 (1.09) | 85.95 (2.30) | 93.79 (0.25) | 93.15 (0.33) | 278.52 (1.28) |
| ER-FIFO-RW-PL-3-1 | 89.19 (0.43) | 90.03 (0.40) | 87.56 (1.09) | 93.68 (0.15) | 92.88 (0.38) | 274.58 (0.62) |
| ER-FIFO-RW-PL-10-1 | 89.84 (0.54) | 90.56 (0.49) | 88.95 (1.30) | 93.39 (0.31) | 91.80 (0.59) | 280.12 (1.85) |

Table 18: Results on Poverty-WPDS ($\alpha = 50\%$). Results are Pearson correlations.

| Algorithm | Avg Online | Avg Recent | Worst Recent | Avg Reg | Worst Reg | Avg Time |
|---|---|---|---|---|---|---|
| FBO | 0.781 (0.024) | | | | 0.638 (0.017) | |
| i.i.d. offline | 0.832 (0.010) | | | | | |
| NBO-100 | 0.813 (0.003) | 0.831 (0.004) | 0.780 (0.010) | 0.583 (0.012) | 0.520 (0.035) | 204.69 (2.12) |
| ER-FIFO-50 | 0.791 (0.007) | 0.806 (0.008) | 0.748 (0.050) | 0.633 (0.018) | 0.549 (0.109) | 881.86 (57.91) |
| ER-FIFO-RW-50 | 0.793 (0.006) | 0.807 (0.009) | 0.753 (0.028) | 0.641 (0.016) | 0.612 (0.024) | 244.83 (3.85) |
| ER-FIFO-RW-100 | 0.798 (0.007) | 0.812 (0.009) | 0.765 (0.045) | 0.641 (0.018) | 0.611 (0.032) | 426.77 (2.04) |
| ER-FIFO-RW-150 | 0.799 (0.004) | 0.813 (0.007) | 0.765 (0.044) | 0.643 (0.015) | 0.608 (0.021) | 629.51 (8.50) |
| MIR-30-4 | 0.737 (0.021) | 0.759 (0.016) | 0.691 (0.050) | 0.579 (0.015) | 0.502 (0.036) | 256.05 (31.52) |
| MIR-100-4 | 0.750 (0.007) | 0.773 (0.015) | 0.702 (0.023) | 0.585 (0.007) | 0.524 (0.043) | 847.82 (27.40) |
| MIR-30-10 | 0.759 (0.010) | 0.779 (0.016) | 0.694 (0.060) | 0.602 (0.008) | 0.561 (0.019) | 719.60 (42.89) |
| MIR-100-10 | 0.774 (0.007) | 0.793 (0.011) | 0.747 (0.029) | 0.611 (0.014) | 0.568 (0.031) | 2148.85 (137.06) |
| MaxLoss-30-4 | 0.727 (0.023) | 0.752 (0.020) | 0.688 (0.038) | 0.575 (0.014) | 0.509 (0.024) | 274.64 (25.15) |
| MaxLoss-100-4 | 0.740 (0.019) | 0.760 (0.023) | 0.677 (0.039) | 0.584 (0.017) | 0.471 (0.182) | 818.20 (19.62) |
| MaxLoss-30-10 | 0.754 (0.012) | 0.776 (0.016) | 0.695 (0.053) | 0.601 (0.008) | 0.561 (0.015) | 557.77 (23.40) |
| MaxLoss-100-10 | 0.770 (0.010) | 0.790 (0.015) | 0.734 (0.054) | 0.610 (0.017) | 0.569 (0.029) | 1643.23 (116.68) |
| GEM-PDS-30-4 | 0.771 (0.005) | 0.800 (0.009) | 0.783 (0.009) | 0.634 (0.014) | 0.622 (0.015) | 338.23 (7.42) |
| GEM-PDS-100-4 | 0.777 (0.006) | 0.809 (0.005) | 0.793 (0.013) | 0.640 (0.010) | 0.627 (0.012) | 1125.91 (36.97) |
| GEM-PDS-30-10 | 0.761 (0.004) | 0.786 (0.006) | 0.773 (0.013) | 0.634 (0.014) | 0.622 (0.012) | 750.12 (4.47) |
| GEM-PDS-100-10 | 0.767 (0.005) | 0.798 (0.006) | 0.781 (0.012) | 0.640 (0.009) | 0.629 (0.011) | 3247.20 (472.93) |
| L2Reg-100-0.1 | 0.797 (0.007) | 0.804 (0.004) | 0.743 (0.027) | 0.614 (0.012) | 0.559 (0.032) | 249.46 (19.51) |
| L2Reg-100-1.0 | 0.792 (0.006) | 0.807 (0.004) | 0.783 (0.008) | 0.642 (0.005) | 0.619 (0.010) | 247.66 (6.89) |
| L2Reg-100-10.0 | 0.780 (0.010) | 0.799 (0.006) | 0.778 (0.013) | 0.641 (0.004) | 0.616 (0.008) | 247.39 (7.83) |
| EWC-100-0.1 | 0.726 (0.009) | 0.744 (0.012) | 0.719 (0.018) | 0.625 (0.010) | 0.601 (0.013) | 884.09 (12.90) |
| EWC-100-1.0 | 0.735 (0.004) | 0.753 (0.007) | 0.734 (0.007) | 0.648 (0.003) | 0.628 (0.007) | 714.14 (61.72) |
| EWC-100-10.0 | 0.716 (0.004) | 0.731 (0.007) | 0.718 (0.006) | 0.652 (0.006) | 0.636 (0.015) | 719.71 (66.79) |

