# OpenReview forum: "Online Continual Learning for Progressive Distribution Shift (OCL-PDS): A Practitioner's Perspective"
_ICLR.cc/2023/Conference — Submitted to ICLR 2023_

### Official Review · Reviewer_UPWo · 2022-10-17

**Confidence:** 5
**Correctness:** 3
**Technical Novelty And Significance:** 2
**Empirical Novelty And Significance:** 2
**Recommendation:** 5

**Clarity, Quality, Novelty And Reproducibility:**

I have covered all these aspects in the above section. Therefore, I use this part of the review to clarify the remainder of the questions for the paper:
* Since the paper assumes infinite storage, what was the buffer size for replay during continual learning?
* How was the random label feedback adopted in the evaluation? Does the paper consider the fraction of users for evaluating the model output?
* The paper experiments with pseudo-labeling semi-supervised methods; was there any specific reason to consider these methods instead of using self-supervised methods that do not require labels?


**Strength And Weaknesses:**

I am well-familiar with the literature and read the full paper in detail. Accordingly, now I will describe the strengths and weaknesses of the paper in the order of originality + significance, quality of the paper, and clarity.

### Originality
##### Strengths
* The paper takes an important step to develop CL benchmarks based on the issues faced by the practitioners. The proposed benchmarks model the shifts in hot topics, time, and language use and would be of interest to the CL community.
* It provides various insights into the existing methods in the proposed OCL-PDS setting. In Particular, observation two, three, and five would be valuable for developing future methods in this setting.

##### Weaknesses
The paper lacks the positioning and comparison of the proposed benchmarks and observations with many prior works:
* **Proposed benchmarks:** The proposed benchmarks are a direct extension of the WILDS benchmarks. Cai et al. [1] proposed Continual LOCalization (CLOC) – 39 million images over nine years, with 712 classes. The paper refers to Lin et al. [2], which introduced CLEAR with 7.8M images based on the temporal evolution of visual concepts of Internet images in the footnote; however, it does not clarify the challenges in the creation of the regression set for CLEAR following the proposed approach. Both these benchmarks have a much larger scale than the benchmarks proposed in the paper. I would also suggest comparing the CL video benchmarks in the paper benchmarks section, providing the limitations of these already existing benchmarks and the necessity for the new benchmarks.
* **Supervised and unsupervised methods:** The paper does not compare or refer to current online CL methods [3,4]. Further, while the paper does not consider many recent unsupervised/self-supervised CL methods [5,6].

---

### Quality
The paper evaluates multiple benchmarks that reciprocate real-world distribution shifts. It focuses on both the supervised and semi-supervised settings, strengthening the overall experimental evaluation. However, I have various concerns regarding the modifications and the proposed setup that I highlight below:
* The assumption of infinite storage is impractical for most applications. Especially in the online setting, where a large number of instances are collected every second, it is not feasible to store the complete dataset. Furthermore, data storage is often restricted for many applications due to privacy restrictions.
* The choice of the time window for knowledge retention needs to be clarified. There are no guidelines to select this hyper-parameter, and it will lead to the formulation of methods that can only remember the recent knowledge, where “recent” is artificially curated for each benchmark. Further, the importance/critical data for the regression set is not well-defined. It is also unclear whether the regression set should change or update depending on the change in the criticality of the data.

---

### Clarity
The paper was overall clear and well-written. Following are a few suggestions that highlight the missing details and should help improve the overall presentation:
* ***Details about regressions set:*** The selection of the regression set should be elaborated in the paper. For instance, for FMoW-WPDS and Amazon-WPDS, how is the regression set constructed? Does it contain all the samples from America, Asia, and ten product categories for FMoW-WPDS and Amazon-WPDS, respectively?
* ***Clarity of the figures:*** The figures for the experimental results are not interpretable. For instance, in Figure 2, each point corresponds to one algorithm according to the caption. However, the figure has no labels, and it is challenging to analyze the performance of any method in the figure. The tables in the supplementary material are much easier to read than the figures; however, the notations could be more straightforward and concise.

---

### Reproducibility
The paper included the code in the supplementary, which would promote the usage of these benchmarks for future CL papers.

---

### References
[1] Cai et al. Online Continual Learning with Natural Distribution Shifts: An Empirical Study with Visual Data. ICCV 21.
[2] Lin et al. The CLEAR Benchmark: Continual LEArning on Real-World Imagery.  NeurIPS 21.
[3] Yin et al. Mitigating Forgetting in Online Continual Learning with Neuron Calibration. NeurIPS 21.
[4] Yoon et al. Online Coreset Selection for Rehearsal-based Continual Learning. ICLR 22.
[5] Madaan et al. Representational Continuity for Unsupervised Continual Learning. ICLR 22.
[6] Fini et al. Self-Supervised Models are Continual Learners. CVPR 22.


**Summary Of The Paper:**

* The paper attempts to bridge the gap between the continual learning settings used in academic work and industrial applications. It introduces the setting of Online Continual learning for Progressive Distribution Shift (OCL-PDS) that considers subtle, gradual, and continuous distribution shifts.
* The paper proposes three modifications to existing CL settings:
  * Online evaluation and training.
  *  Remember recent and important knowledge described by regression set instead of the complete past knowledge.
  * Infinitely large storage without the replay of all samples.
* The evaluation is conducted on four benchmarks proposed for this setting based on the WILDS benchmark for supervised and semi-supervised settings.


**Summary Of The Review:**

While the paper is interesting and tackles a significant problem, it is not ready for acceptance in its current form. Remarkably, the paper lacks a comparison with existing benchmarks and CL methods, the modifications to the CL settings could be better-motivated and appropriate for many practical scenarios, and the paper's clarity (especially the results) needs to be significantly improved.

---

> ### Author Response · Authors · 2022-11-08
> **Response to Reviewer UPWo (1/2)**
>
> Thank you for your very detailed comments! We would like to address your concerns as follows:
> 1. Regarding CLOC and CLEAR: First we would like to sincerely apologize for not citing CLOC which is a closely related paper - We did have a comparison with Cai et al. in the paper before but accidentally removed it when we were fitting the main text into nine pages. We have added the removed part back in the revision, and an additional full comparison with this work in Appendix A.2.1. Both CLOC and CLEAR are PDS benchmarks and can be used for OCL-PDS. While they do not have an intuitive definition of the regression set as our benchmarks do, one can still artificially define regression sets on these two benchmarks. However, both CLOC and CLEAR are image classification tasks, while our benchmarks cover a wider range of tasks. Moreover, as we mentioned in Section 6, there are definitely a lot of other exciting applications not covered in our benchmarks, and the four benchmarks in this paper are just a starting point. In future work, we will adapt more datasets, such as those for videos, robotics, time series and so on, to OCL-PDS, which hopefully will even attract more people to work on this important and exciting problem, and develop better OCL algorithms.
> 2. Regarding not comparing recent supervised/semi-supervised OCL methods: We thank the reviewer for bringing these papers up, and have added them to our citations. The reason why we did not compare these methods is that Continual Learning is a very big and flourishing field, and there are a lot of new papers proposing new methods each year. With that said, we cannot compare all methods in a single paper because doing so would make the results very hard to read and process. Therefore, we carefully chose a limited number of CL methods to compare in the paper, and we only chose the classical ones that represent the typical and most popular CL strategies. However, we do make our benchmarks super easy to use (see the readme in our code), so it is very easy to run new methods on our benchmarks. Moreover, we plan to create a website with a leaderboard to compare new OCL methods, and people can rank, select and compare different methods with a few clicks, visualize the results, analyze the effects of hyperparameters, etc. This would be a better way of representing these results than putting all methods in a large table which would be quite hard to read.
> 3. Regarding the infinite storage assumption: This is a very good point and we would like to clarify. We agree with the reviewer that for some applications with a huge amount of data, even for the industry it is not practical to store all the samples. However, we want to clarify that “infinite storage” is only an *assumption*, and the reason why we made this over-optimistic assumption is that we want to focus on more practically relevant questions than the storage limit. As we wrote in the paper, the storage size is rarely the bottleneck for industrial applications, so we do not need to “put too much effort into utilizing every bit of storage”. Instead, we would like to focus on more important questions, such as how to leverage the unlabeled data better, how to make the OCL methods more efficient, how to balance between the online performance and the knowledge retention performance, etc. Moreover, even though we assume infinite storage, we do not replay all previous samples during OCL because it would be too inefficient. We thank the reviewer for this comment and have made clarifications on this point in the revision page 4.
> 4. Regarding some applications having privacy restrictions: This is a good point. We definitely agree with the reviewer and have touched on this point in Section 2.3. In fact, different industrial applications have different restrictions - some might have privacy restrictions, and others such as end devices might have local memory or network bandwidth/latency  restrictions. As we wrote in Section 6, “it is very difficult to cover all the tasks within a single general problem formulation”. For any specific application, we still need to carefully consider its characteristics and subtleties. However, a general formulation like OCL-PDS is still very useful because it can help researchers develop better general algorithms which can then be applied to specific applications with careful adaptations.
> 5. Regarding the choice of the time window: For real-world applications, the time window depends on the specific application. As we mentioned in Section 2.2, the point of considering the recent knowledge retention performance is that the distributions might appear in a recurring fashion. For example, for a doorbell camera, the images in the morning and at night look very different, but the images in two consecutive mornings are very similar. In this case, the time window could be set to one day.

---

> > ### Author Response · Authors · 2022-11-08
> > **Response to Reviewer UPWo (2/2)**
> >
> > 6. Regarding the definition of the regression data: First we would like to emphasize that the concept of the regression set is very common in industrial applications, including the ones we and our fellow practitioners are working on. The regression data is well defined in those real applications, and the frequent data and the critical data are the two most common definitions (we note that what is considered “critical” depends on the concrete industry application). Two important characteristics of regression data are: (i) Its labeling function does not change or changes very little (no concept shift); (ii) It has a bigger weight in the overall loss (more costly to make mistakes on). We thank the reviewer for this comment and have edited our description of the regression set in the revision page 3.
> > 7. Regarding the “change in the criticality of the data”: This is very rare in practice, and when such a change happens, practitioners would usually retrain the model. One example we could think of is fairness: Several years ago people found that existing models were biased towards certain minority groups, so the criticality of the data changed and practitioners retrained the models to address this fairness issue. This, however, is a rare event and is beyond the scope of this work.
> > 8. Regarding Figure 2: We agree with the reviewer that one cannot find the performance of any individual method from Figure 2. However, the reason why we did not label the points in Figure 2 is that doing so would make the figure very cumbersome and very hard  to read. Meanwhile, even without the labels, we can get Observations #1 and #2 from Figure 2. Also, we do have detailed results in the tables in the appendix. As we said earlier, we plan to create a website with result visualization where one can click on the points to see the performance of each individual method, which would be a much better way to represent these results.
> > 9. Regarding the replay buffer size: Even though we assume infinite storage, we do not replay all previous samples during OCL because that would be too slow, and we do consider training efficiency in our evaluation metrics. We can define the “essential replay buffer size” as the number of previous samples replayed for each time step. In our experiments, we keep the essential replay buffer size linear in the size of the new data batch. For example, in ER-FIFO-RW, it is twice the number of new labeled samples, meaning that the method will take twice more time than training on the new labeled samples alone.
> > 10. Regarding the random label feedback: The random label feedback is adopted by using a semi-supervised setting in the OCL procedure. $\alpha$ is the number of labeled samples divided by the total number of samples.
> > 11. Regarding training without labels: When $\alpha = 0$, this is the unsupervised learning setting where there are no labels from batches after t = 1. This setting, however, is not very useful for two reasons: (i) As we mentioned in the last paragraph of Section 5, if there is concept shift such that $P(Y|X)$ changes, then there is no hope for the model to learn this knowledge without labels, in which case unsupervised OCL is infeasible; (ii) Even if there is no concept shift, the Wilds paper [1] found that existing methods for distribution shift “yield substantially lower out-of-distribution than in-distribution performance” without any labels from the target distributions. In fact, we did run pseudo-labeling and FixMatch for $\alpha = 0$ and the result is very poor, similar to Observation #5 where there is no virtual update. Moreover, from a practical point of view, in real applications it is rarely the case that we cannot obtain any labels from the target distribution. There are always some ways to collect new labels, such as human annotation and user error reports.
> >
> > We again thank the reviewer for this very detailed and useful feedback, and hope that our response addresses the raised concerns. We sincerely hope that the reviewer could reconsider the novelty and significance of this work, and how this work, especially the OCL-PDS problem formulation, could help boost the development of better OCL algorithms as well as inspire more researchers and practitioners to study real-world PDS. We are looking forward to having further discussions with you during the discussion period.
> >
> > [1] Koh et al., WILDS: A Benchmark of in-the-Wild Distribution Shifts, ICML 2021.

---

### Official Review · Reviewer_6758 · 2022-10-23

**Confidence:** 4
**Correctness:** 3
**Technical Novelty And Significance:** 2
**Empirical Novelty And Significance:** 2
**Recommendation:** 3

**Clarity, Quality, Novelty And Reproducibility:**

* Clarity: overall well written, but as a paper introducing benchmark datasets, the procedure on how the datasets are constructed is only briefly discussed, and the experiments part can be improved, for example, figure titles are not informative at all.
* Quality: fair
* novelty: the novelty is limited, the problem is not not new, datasets are not new, algorithms are not new, missing citation of similar existing work
* reproducibility: appears to be good


**Strength And Weaknesses:**

* The paper tries to be practical and bridge the gap between academic and industry, but I don’t really see anything novel. Missing citation: Cai, Sener, Koltun, “Online Continual Learning with Natural Distribution Shifts: An Empirical Study with Visual Data”, ICCV 2021.
* An important contribution is the benchmark datasets, and a 3-step procedure is described, but no examples are shown in the main text how these 3 steps are executed, and how the distribution is shifting with some metrics to demonstrate. I’m not very convinced these benchmark dataset are good.
* Most of the results are presented in the format of scatter plot of “worst reg perf” vs  “avg online perf”, but it’s not well motivated why this is a good way of demonstrating the benchmarks are good.


**Summary Of The Paper:**

This paper is intended to formally introduce the online continual learning problem with progressive distribution shift, and also presented four benchmark datasets and 12 algorithms (adapted from existing algorithms) for this setting, and finally presented empirical results of these algorithms on these datasets.

**Summary Of The Review:**

This paper attempts to establish some benchmark datasets and algorithms for online continual learning under progressive distribution shift. While a practitioners’ perspective is emphasized, I failed to see much novelty of this perspective, and the empirical results are not presented in a convincing enough way to show the benchmarks are good enough for wide use for the community.

I would recommend the authors to check out this paper and explain the differences: Cai, Sener, Koltun, “Online Continual Learning with Natural Distribution Shifts: An Empirical Study with Visual Data”, ICCV 2021.

---

> ### Author Response · Authors · 2022-11-08
> **Response to Reviewer 6758 (1/2)**
>
> Thank you for your very detailed comments! We would like to address your concerns as follows:
> ### Regarding the comparison with Cai et al.
> First we would like to sincerely apologize for not citing Cai et al. which is a closely related paper - We did have a comparison with Cai et al. in the paper before but accidentally removed it when we were fitting the main text into nine pages. We have added the removed part back in the revision, and an additional full comparison with this work in Appendix A.2.1. Here we would like to describe the differences between these two papers as follows:
> 1. Cai et al. considered a fully supervised setting: The environment reveals all true labels to the learner after evaluation. In contrast, in OCL-PDS, we use the random label feedback which randomly selects $\alpha$ fraction of the new samples and provides their labels. In particular, we mainly study a semi-supervised learning setting where the environment only reveals a fraction of the labels, which is a more common scenario in industrial applications.
> 2. The evaluation metrics used in Cai et al. were: Average online accuracy, backward transfer and forward transfer. The first two metrics correspond to the average online performance and the average recent knowledge retention in OCL-PDS. In addition to these metrics, OCL-PDS also considers the regression set performance which is a very important metric in real applications, the worst (recent/important) knowledge retention performances since knowledge retention is required for all t, and the training efficiency which is very important in an online setting where the algorithm is run for many times.
> 3. Cai et al. only evaluates knowledge retention performances at three fixed time steps: H/3, 2H/3 and H, where H is the total number of time steps. In contrast, OCL-PDS is **purely online**: all metrics including knowledge retention are evaluated at each step.
> 4. Cai et al. proposed the CLOC benchmark which is a PDS benchmark without a definition of the regression set. However, CLOC only covers vision classification, while our benchmarks cover both language and vision tasks, and both classification and regression tasks, which is more comprehensive.
> 5. Cai et al. only run experience replay (ER) on their benchmark, while our work also compares regularization based methods such as EWC, variants of ER such as GEM, MIR and MaxLoss, as well as semi-supervised learning methods like pseudo-labeling and FixMatch.

---

> > ### Author Response · Authors · 2022-11-08
> > **Response to Reviewer 6758 (2/2)**
> >
> > ### Regarding the novelty of this work
> > We would like to reiterate the novelty of this work as follows:
> > 1. The OCL-PDS problem formulation, as detailed in Section 2.1, is novel. The three major novelties of this new setting are described in the introduction (Section 1). It is different from the setting studied in Cai et al. as we pointed out in the previous point. Based on our literature review and our first-hand experience on real applications, we believe that this new formulation is closer to industrial applications and is better aligned with practitioners' needs, which we argued in detail in Section 2.3. We believe that this new formulation can help researchers focus on the more important aspects in OCL, and hope that it can inspire more people to work on this important and exciting problem.
> > 2. The 3-step benchmarking procedure and the 4 benchmarks we build in Section 3 are novel. It is true that the datasets we use are existing ones. However, in our benchmarks, we are using these datasets in a substantially different way than how they have been used previously, so the benchmarks should be considered new. Besides, we describe the 3-step procedure we used to construct these 4 benchmarks in Section 3.1 and demonstrate it in detail in Appendix C, and this procedure can be used to construct new PDS benchmarks on datasets from other domains (e.g. robotics), so we believe that this benchmarking procedure is also an important contribution.
> > 3. Although the algorithms we compare in the paper are not new, we still need to adapt these algorithms to the new problem formulation studied in this paper. For example, GEM-PDS is a combination of GEM and A-GEM adapted to OCL-PDS.
> > 4. We make some new observations in our experiments. For example, observation #3 shows that many existing CL algorithms cannot do well on regression tasks, and observation #5 reveals the importance of virtual update in semi-supervised continual learning which was not discovered in previous work such as gradual self-training.
> >
> > We sincerely hope that the reviewer could reconsider the novelty of this work, and how the contributions of this work could help boost the development of better OCL algorithms as well as inspire more researchers and practitioners to study this problem. We would be very happy to answer any further questions the reviewer might have.
> >
> > ### Regarding the 3-step benchmarking procedure
> > We agree with the reviewer that the 3-step benchmarking procedure is important, so we have added a brief description of how the 3 steps are executed in Section 3.1. The more detailed description, which is rather long,  can be found in Appendix C.1. We hope that the reviewer agrees with our decision to move the detailed description to the appendix as it cannot be adequately described in the main text due to the page limit.Regarding the experiment results: Full experiment results can be found in Appendix E. In the main text, we use the scatter plots of average online performance vs worst regression set performance to demonstrate our observations.
> >
> > We again thank the reviewer for this very detailed and useful feedback, and sincerely hope that our response can address your concerns. We are looking forward to having further discussions with you during the discussion period.

---

### Official Review · Reviewer_FcFA · 2022-10-24

**Confidence:** 4
**Correctness:** 3
**Technical Novelty And Significance:** 4
**Empirical Novelty And Significance:** 4
**Recommendation:** 10

**Clarity, Quality, Novelty And Reproducibility:**

Clarity: Good

Quality: Good

Novelty: Good

Reproducibility: Good

**Strength And Weaknesses:**

Strengths:
1. This work first introduces and investigates the novel OCL-PDS problem, which more closely aligns with practitioners’ needs in the industry;
2. This work improves the setting of conventional DA and CL problems at three points, including task-free, forgetting-allowed, and infinite-storage, which is reasonable and more practical, and thus can close the gap between academic work and real industrial applications to some degree.
3. This work requires only remembering the “recent knowledge” and “important knowledge” rather than the knowledge on all tasks and designs a “regression set” to describe the important knowledge.
4. This work releases 4 new benchmarks using the 3-step procedure for the OCL-PDS setting, while adapting and implementing 12 OCL algorithms and baselines, including both supervised and semi-supervised. Their observations based on extensive experiments may help boost the development of OCL algorithms for handling PDS.

Weaknesses:
1. The “infinite storage” this work proposes is somewhat inconsistent with the actual situation, especially in scenarios where the distribution changes gradually over time, such as a recommendation system. The space cost that increases linearly with time T will bring a heavy burden when T becomes large even for the industrial scenarios.
2. According to observation 2, the online performance depends on how the regression set is defined, and how close it is between the regression set distribution and the overall distribution. So besides the proposed “frequent data” and “critical data”, is there a more general way to construct an appropriate regression set to describe the important knowledge?
3. It is expected to add First Batch Only (FBO) without the training regression set as a baseline, which can serve as a better lower bound of the online performance than FBO. Because the critical data may not show in the first batch.
4. The proposed i.i.d. offline may not be the upper bound of the online performance, because jointly learning multiple datasets may outperform learning each dataset independently.


**Summary Of The Paper:**

This paper introduces a novel “Online Continual Learning for Progressive Distribution Shift (OCL-PDS)” problem that widely exists in industrial applications from the practitioner’s perspective, aiming to close the gap between academic research and industry. They propose three modifications to the related conventional settings, build 4 new benchmarks from the Wilds dataset, and implement 12 algorithms and baselines including both supervised and semi-supervised methods. Extensive experiments on the new benchmarks also bring some observations.

**Summary Of The Review:**

The proposed setting, concepts, evaluation paradigms, and datasets are realistic and may have a large impact on industrial applications.

---

> ### Author Response · Authors · 2022-11-08
> **Response to Reviewer FcFA**
>
> Thank you for your very detailed comments! We would like to address your concerns as follows:
> 1. Regarding the infinite storage assumption: This is a very good point and we would like to clarify. We agree with the reviewer that for some applications with a huge amount of data, even for the industry it is not practical to store all the samples. However, we want to clarify that “infinite storage” is only an *assumption*, and the reason why we made this over-optimistic assumption is that we want to focus on more important questions than the storage limit. As we wrote in the paper, the storage size is rarely the bottleneck for industrial applications, so we do not need to “put too much effort into utilizing every bit of storage”. Instead, we would like to focus on more practically relevant questions, such as how to leverage the unlabeled data better, how to make the OCL methods more efficient, how to balance between the online performance and the knowledge retention performance, etc. Moreover, even though we assume infinite storage, we do not replay all previous samples during OCL because it would be too inefficient. We thank the reviewer for this comment and have made clarifications on this point in the revision page 4.
> 2. Regarding “a more general way to construct an appropriate regression set”: For any real application, the definition of the regression set depends on the application itself. The frequent data and the critical data are two very common types of regression data that appear in many real applications, including the ones we are focusing on. Two important characteristics of the regression data are: (i) Its labeling function does not change or changes very little (no concept shift); (ii) It has a bigger weight in the overall loss (more costly to make mistakes on). We thank the reviewer for this comment and have edited our description of the regression set in the revision. A general way of representing the regression data can be a binary classifier F(x), such that F(x) = 1 indicating that x belongs to the regression set. If we have access to this F(x), then we can use methods such as importance weighting to make the model focus more on the regression data. This by itself is a very interesting topic, and we leave it to future work.
> 3. Regarding “First Batch Only without the training regression set”: We want to clarify that the training regression set is a part of the initial training set (which consists of the first batch and the training regression set). An easy way of thinking it is this: The initial training set contains an extra binary label indicating whether each sample is in the regression set or not. The initial training set is the same for all algorithms and baselines (i.e. all methods receive the training regression set in the first epoch), so FBO serves as a valid lower bound. We have added clarifications in the revision to address this confusion.
> 4. Regarding “i.i.d. offline may not be the upper bound of the online performance”: This is a very good point. We agree with the reviewer that in some cases, training on multiple distributions can achieve a higher performance on distribution $D_t$ than training on $D_t$ alone, so i.i.d. offline is not necessarily an upper bound of the online performance for all methods. To address this issue, as described in Appendix D, we make sure that the training set size of i.i.d. offline is sufficiently large (at least the size of the union of previous batches). The point of this baseline is to take the generalization gap into account, and an “approximate” upper bound can help us interpret the performances of OCL algorithms better. We thank the reviewer for this comment and have made clarifications on this point in the revision Appendix D.
>
> We again thank the reviewer for this very detailed and useful feedback, and sincerely hope that our response can address your concerns. We are looking forward to having further discussions with you during the discussion period.

---

### Official Review · Reviewer_FxrK · 2022-10-28

**Confidence:** 4
**Correctness:** 3
**Technical Novelty And Significance:** 3
**Empirical Novelty And Significance:** 3
**Recommendation:** 6

**Clarity, Quality, Novelty And Reproducibility:**

Paper is clearly written. There is novelty but author's claim that their method is more practical is not justified.

**Strength And Weaknesses:**

Strengths:
1) Paper is clearly written and the authors point to practical issues with the current approaches.
2) Authors give very detailed justification of their choices, give details of experiments which will be helpful for other researchers to build on.

Weaknesses:
1) The proposed approach still has some practical issues (e.g. how does one decide the divergence threshold, how does one make sure that 3-step procedure benchmark OCL-PDS is applicable to real world problems?
2) The final takeaway is not clear. Authors should discuss some of the real world problems and give some conclusions based on it. It is not clear what part of the discussion is helpful if a real world practitioner wants to use any of the knowledge in the paper.


**Summary Of The Paper:**

Authors introduce the novel OCL-PDS problem - Online Continual Learning for Progressive Distribution Shift. Authors contrast their problem with that of continual learning and domain adaptation. Authors build 4 new benchmarks and implement 12 algorithms to test these benchmarks.


**Summary Of The Review:**


1) From Appendix B, one can understand why divergence should be asymmetric. But from the example given, it looks like one needs structured asymmetry. Is just having asymmetry enough?

2) Figure 1: It is not clear what distribution shift we are talking about in "FMoW-WPDS benchmark." Are most images in 2002 of prison? Are most images in 2009 of Helipads?

3) Section 2.3: Work is also very related to domain generalization (DG) which is a harder problem compared to domain adaptation (DA) [1]. If one wants to be truly "online" then one DG might be a better setting than DA? Authors should add more discussion comparing and contrasting with DG too.

4) In general, real world datasets do not have shift continuity. Or there could be too much non-stationarity. How would ODD check or shift continuity check work in that case?

5) How does one decide threshold on Div (D_t || D_t+1)? In the real world dataset, this could be even harder to determine.

6) I checked the hyperparameters considered In Table 14(in appendix). But the list of hyperparameters tuned does not look complete. The training data generation (domains, dist shift and threshold on div for continual shift) should be treated as hyperparameters too. Can authors comment more on this?




[1] Blanchard, Gilles, Aniket Anand Deshmukh, Ürun Dogan, Gyemin Lee, and Clayton Scott. "Domain generalization by marginal transfer learning." The Journal of Machine Learning Research 22, no. 1 (2021): 46-100.

---

> ### Author Response · Authors · 2022-11-08
> **Response to Reviewer FxrK (1/2)**
>
> Thank you for your very detailed comments! We would like to address your concerns as follows:
> 1. Regarding “how does one decide the divergence threshold, how does one make sure that 3-step procedure benchmark OCL-PDS is applicable to real world problems”: We would like to make this important clarification that the point of this 3-step benchmark procedure is to simulate PDS on existing datasets which have not been benchmarked for PDS. In real-world problems that naturally contain PDS, there is no need to decide the threshold or to use any benchmarking procedure because the data already has them. Our problem formulation and benchmarking process are applicable to most real-world problems, but there are still some real-world problems which they cannot cover, because as we wrote in Section 6, “the application of deep learning has become so wide nowadays that it is very difficult to cover all the tasks with a single general problem formulation”. For any individual real-world task, we still need to carefully consider the characteristics and subtleties of the task, and tailor our algorithms towards them. However, a general problem formulation and a general benchmarking procedure are still very useful for the development of better algorithms and comparison among methods. Moreover, the shift continuity check described in Section 3.1 is used to decide the divergence threshold in the benchmarking procedure.
> 2. Regarding the final takeaway: We believe that the most important takeaway from this work is the new OCL-PDS problem formulation - We discovered a wide gap between previous formulations and real-world applications through our literature review and conversations with industry practitioners, and in Section 2.3 we discussed in detail how the new OCL-PDS formulation closes this gap. Hopefully this work could inspire more researchers to work on this important and exciting problem. Other deliverables of this work include: 4 new benchmarks for OCL-PDS, a 3-step benchmarking procedure so that follow-up work can create benchmarks on new domains (e.g. robotics), the implementation of 12 baselines and algorithms, and a comparison of these algorithms with insights into existing methods (Observations #1 - #5). The benchmarks and algorithms we provide are by design super easy to use (see the readme in the code), and they could be very useful to industry practitioners to test their OCL algorithms.
> 3. Regarding the “author's claim that their method is more practical is not justified”: In the paper the only two things we claim as “more practical” are: (i) our new OCL-PDS problem formulation is more practical than previous problem settings, which we have discussed and provided detailed justification for in Section 2.3; (ii) Our benchmarks are more practical than the three most common types of benchmarks widely used in the CL literature, which we have justified in the beginning of Section 3.
> 4. Regarding “structured asymmetry”: As elaborated in Appendix B, the reason why we need asymmetry in the divergence function is that we want $Div(P \parallel Q)$ to reflect the performance of a model trained on P and tested on Q. We can use structured asymmetry into the divergence function if it is beneficial for this purpose, but at this point we cannot see which kind of structured asymmetry helps. If the reviewer has an idea on this, we will be more than happy to discuss further.
> 5. Regarding the FMoW-WPDS benchmark: Each year of data contains samples from all 62 classes. There is no major class shift over the years in this dataset, and we are just presenting random samples from 2002, 2009 and 2016 in the original Figure 1(a). We have updated Figure 1(a) to address this confusion.
> 6. Regarding domain generalization: This is a good point. First, in OCL-PDS, the model is tested on the new data batch before it is fine-tuned on the new batch, so essentially the model is tested on a new distribution from which it never saw any samples before, which is to some extent similar to “online domain generalization” (though the domain shift in OCL-PDS is much smaller than that in conventional DG). Second, regarding comparison with DG, DG is the scenario where no samples (even unlabeled ones) from the new domain are provided, but in industrial applications it is extremely rare that we can get no samples from the new domain at all, as we argued in Section 2.3 (PDS vs DA/DG). We have added the citation the reviewer mentioned.

---

> > ### Author Response · Authors · 2022-11-08
> > **Response to Reviewer FxrK (2/2)**
> >
> > 7. Regarding “how would OOD check or shift continuity check work when real world datasets do not have shift continuity”: How shift continuity check works is described in detail in Appendix C. Basically, we train a model on a training set of samples from previous distributions, and then test the model on a validation set of samples from previous distributions, as well as a test set of samples from the new distribution. If the gap between the validation set performance and the test set performance is small, then the shift is considered continuous. We also want to clarify for existing datasets that have not been benchmarked for OCL-PDS yet, we need to first benchmark OCL-PDS with the three-step procedure, and the shift continuity check is performed between Step 2 and Step 3.
> > 8. Regarding “how does one decide the threshold on $Div (D_t \parallel D_{t+1})$”: We have answered this question in point 1. For a real-world application with real PDS, there is no need to decide this threshold - The task naturally sets the threshold. In our benchmarking procedure, we perform the shift continuity check to make sure that the shift is continuous and not too big.
> > 9. Regarding benchmark hyperparameters: This is a good point and we thank the reviewer for bringing this point up. Indeed, it would be useful to see how these OCL algorithms perform under different divergence magnitudes, labeled data fraction, etc. We have done experiments with different choices of $\alpha$ (the labeled data fraction): For instance, we tried both $\alpha = 0.5\%$ and $\alpha = 5\%$ for CivilComments-WPDS. The online performances go up with more labeled samples, but the observations are the same as Section 5. We are running these extra experiments right now and will release more results with different benchmark hyperparameters in future edits of this paper.
> >
> > We again thank the reviewer for this very detailed and useful feedback, and sincerely hope that our response can address your concerns. We are looking forward to having further discussions with you during the discussion period.

---

> > > ### Comment · Reviewer_FxrK · 2022-12-07
> > > **Response to authors**
> > >
> > > I have read the author's response and reviews from others.
> > > Some of my concerns on divergence threshold, structured asymmetry and domain generalization have been addressed. But some of the concerns are still there. For example, like other reviewers, I don't understand how this work would be widely used and how it can help other researchers. Benchmark hyperparameters comment is still not addressed. Given all the information, I will keep my current score.

---

> > > > ### Author Response · Authors · 2022-12-08
> > > > **Discussion**
> > > >
> > > > Thanks for your response! We are glad that our rebuttal is able to address most of your concerns. For the remaining concerns:
> > > > 1. Regarding whether this work is useful to a wider audience: First, note that only Rev 6758 has expressed similar concern, which we have addressed in our rebuttal. As the title suggests, this work focuses on a practitioner’s perspective, and aims to bridge the gap between current continual learning research and real industrial applications. We firmly believe this work, especially the OCL-PDS formulation, is very useful to industrial practitioners, because it helps them focus on dealing with the right bottlenecks in real applications. And for general researchers, the metrics and benchmarking procedure introduced in this work can help them develop new better OCL algorithms.
> > > > 2. Regarding the benchmark hyperparameters: We acknowledged in the rebuttal that it would be useful to compare the performance of OCL under different hyperparameters. And as we mentioned in our response to other reviewers, we will gradually release new benchmarks in other domains and with larger scales. Our goal is to build a whole suite of PDS benchmarks, which needs to take some time for various reasons (privacy issues, testing, etc.). This work serves as the foundation of this suite of PDS benchmarks because it formulates the OCL-PDS problem and the evaluation procedure of OCL algorithms on a PDS benchmark. We sincerely hope that the reviewer could take this point into consideration.
> > > >
> > > > Again, we would like to thank the reviewer for this very useful discussion. We are happy to answer any further questions from you.

---

### Author Response · Authors · 2022-11-08
**General Response to Reviewers**

We would like to thank all reviewers for their valuable and detailed comments. We have revised the paper according to the feedback and have uploaded the revision, with edits marked in red. We are looking forward to having further discussions with the reviewers before Nov 18.

Thanks,
Paper966 Authors

---

> ### Author Response · Authors · 2022-11-15
> **Discussion Period Ending Soon**
>
> Just a kindly reminder that the discussion period ends on Nov 18. We are looking forward to discussing with the reviewers and are happy to answer any further questions.

---

### Decision · Program_Chairs · 2023-01-20

**Decision:**

Reject

**Justification For Why Not Higher Score:**

There are important flaws identified by the negative reviewers and not addressed by the authors. The paper includes a lot of information but does not seem to have main points.

**Justification For Why Not Lower Score:**

N/A

**Metareview: Summary, Strengths And Weaknesses:**

The paper proposes several benchmarks for evaluating progressive distribution shift (PDS), and evaluated many baseline continual learning (CL) methods on these benchmarks.

The critical reviewers objected to the 3 step construction method being arbitrary, missing references and most importantly that the results are unclear. The authors addressed the missing reference and made further arguments on novelty. As reviewers (FxrK,  6758, UPWo) pointed out, the findings of the paper are unclear, especially on the quantitative side, and the authors did not address this important point sufficiently. For instance, in figure 2, the first two sets have positive correlations as mentioned in observation 1, the next one negative, and the last no correlation. The paper did not take a stand on which CL methods are better on the benchmark or how to better study the problem. It also misses positioning with respect to state-of-the-art CL methods and other related benchmarks (UPWo). The main empirical results are 5 hedged observations. Thus this work does not seem to "provide tools to better handle realistic PDS and design better algorithms".

Note that there is a rating 10 review. The review repeated the paper's main arguments and but did not convey why this is an exciting paper with such a high rating. In fact it pointed out a few problems that other reviewers also pointed to, so this review was treated as a regular positive review as its text indicates.

The paper contains a large appendix ranging from an exposition of various losses to dataset statistics, making the paper difficult to read. The author is encouraged to exercise more editorial judgment to identify and focus on the main points of the paper.



**Summary Of Ac-Reviewer Meeting:**

Only 1 reviewer responded to the doodle. Other reviewers also did not respond to specific email questions from the AC and did not respond to author comments.